# Cyclic AMP induces reversible EPAC1 condensates that regulate histone transcription

Liliana Felicia Iannucci [1,2], Anna Maria D'Erchia [3], Ernesto Picardi [3], Daniela Bettio[4,5], Filippo Conca [1,2], Nicoletta Concetta Surdo[2,6], Giulietta Di Benedetto [2,6], Deborah Musso[1], Cristina Arrigoni[1], Marco Lolicato[1], Mauro Vismara [1,2], Francesca Grisan[2], Leonardo Salviati [4,5], Luciano Milanesi[7], Graziano Pesole [3] & Konstantinos Lefkimmiatis [1,2,6] ✉

The second messenger cyclic AMP regulates many nuclear processes including transcription, pre-mRNA splicing and mitosis. While most functions are attributed to protein kinase A, accumulating evidence suggests that not all nuclear cyclic AMP-dependent effects are mediated by this kinase, implying that other effectors may be involved. Here we explore the nuclear roles of Exchange Protein Activated by cyclic AMP 1. We find that it enters the nucleus where forms reversible biomolecular condensates in response to cyclic AMP. This phenomenon depends on intrinsically disordered regions present at its amino-terminus and is independent of protein kinase A. Finally, we demonstrate that nuclear Exchange Protein Activated by cyclic AMP 1 condensates assemble at genomic loci on chromosome 6 in the proximity of Histone Locus Bodies and promote the transcription of a histone gene cluster. Collectively, our data reveal an unexpected mechanism through which cyclic AMP contributes to nuclear spatial compartmentalization and promotes the transcription of specific genes.

Based on our current understanding most 3′,5′-cyclic adenosine monophosphate (cAMP)-dependent nuclear effects, especially transcription, depend on Protein Kinase A (PKA), a tetrameric cAMP-responsive serine/threonine kinase composed by two regulatory (PKA-Rs) and two catalytic subunits (PKA-Cs)[1,2]. The classic model underlying the nuclear activity of PKA postulates that, in response to cAMP binding, the regulatory subunits of extra-nuclear PKA tetramers undergo conformational changes that release PKA-Cs which can diffuse in the nucleus[3]. On the other hand, eventual nuclear PKA tetramers are reached by cAMP, that can freely enter the nucleus[2], and are activated in situ[4,5]. While the obvious interpreter of nuclear cAMP signals seems PKA, its actions in this compartment are limited by several mechanisms acting at different steps of the cascade. For instance, PKA activation in the nucleus depends on the local cAMP levels that are strictly regulated by phosphodiesterase (PDE)-dependent cAMP hydrolysis[5]. On the other hand, the activity of PKA-Cs in the nucleus is contrasted by protein kinase inhibitors (PKIs), a family of proteins that interact with free PKA-Cs to both, hinder their catalytic activity and vehicle them to the cytosol[6,7]. In addition to these well-established regulatory mechanisms, we and others recently reported that PKA-dependent phosphorylation in the nucleus is strongly inhibited by phosphatases in several cell types[7–11]. These findings are in line

[1]Department of Molecular Medicine, University of Pavia, Pavia, Italy. [2]Veneto Institute of Molecular Medicine, 35129 Padova, Italy. [3]Department of Biosciences, Biotechnologies and Environment, University of Bari "Aldo Moro", Bari, Italy. [4]Clinical Genetics Unit, Department of Women's and Children's Health, University of Padova, Padova, Italy. [5]Fondazione Istituto di Ricerca Pediatrica Città della Speranza, Padova, Italy. [6]Institute of Neuroscience (IN-CNR), National Research Council of Italy, Padova, Italy. [7]Institute of Biomedical Technologies, National Research Council of Italy, Milan, Italy. ✉e-mail: konstantinos.lefkimmiatis@unipv.it

with a number of studies suggesting that PKA is not the sole responsible for all cAMP-driven nuclear functions, but other cAMP effectors are implicated[12–15].

Exchange Proteins Activated by cAMP 1 and 2 (EPAC1 and EPAC2, genes *RAPGEF3* and *RAPGEF4,* respectively) are a family of cAMP-responsive guanine-nucleotide exchange factors (GEFs) able to activate Rap proteins[16,17]. Both EPAC1 & 2 contain structurally similar amino-terminal (N-terminal) regulatory and carboxy-terminal (C-terminal) catalytic regions. Within the regulatory region can be distinguished a Dishevelled, Egl-10 (DEP) domain involved in the subcellular localization of the proteins[12,18] and a cyclic-nucleotide-binding domain (CNBD) responsible for their cAMP-dependent activation[12,19]. While the catalytic region contains a RAS-association domain (RA) and a Ras-like small GTPases domain (Ras-GEF) and a Ras-exchange motif (REM) which is important for the stabilization of EPAC1 & 2 in their active conformation[20]. Despite their structural similarities, the physiological actions of EPACs differ, most likely due to their distinct tissue and subcellular distribution. EPAC2 is mainly expressed in the central nervous system and adrenal gland, and is predominantly found in the cytosol and near the plasma membrane[18]. EPAC1, on the other hand, is ubiquitously expressed[12,21] and is located virtually in all cellular compartments, including the mitochondria[22] and the nuclear envelope[23,24]. EPAC1 was also shown to participate in the regulation of nuclear events, such as the nuclear export of DNA-protein kinase (DNA-PK)[25] and histone deacetylase 4 (HDAC4)[26]. Despite this evidence, the role of EPAC1 in the nucleus as an effector of the cAMP signaling cascade has not been fully elucidated. Here we show that EPAC1 can enter the nucleus and form biomolecular condensates in response to cAMP elevations. EPAC1 condensate formation depends on its N-terminal regulatory region and is independent of PKA activity. Furthermore, we demonstrate that EPAC1 condensates interact with Histone Locus Bodies (HLBs) and affect the transcription of the histone gene cluster 1 at chromosome 6 in a cAMP-dependent manner. Our data provide evidence for a cAMP/EPAC1 axis in the nucleus that affects transcription independently of PKA.

## Results

### EPAC1 enters the nucleus and forms spherical puncta in response to cAMP

EPAC1 can be found soluble in the cytosol but also at the nuclear envelope complexed with the nuclear pore component RAN Binding Protein 2 (RANBP2)[27]. It is assumed that engaging with RANBP2, through its CDC25 homology domain, is necessary and sufficient for the entry of EPAC1 in the nucleus[23,28], however, the existence of other domains that may participate in this process has never been tested. We performed in silico analysis using the algorithm NLS Mapper[29] and identified two putative nuclear localization sequences (NLS) within EPAC1 (Supplementary Fig. 1a), namely amino acids 732–764 and 179–208. One of these (AAs 732–764) partially overlapped with the nuclear pore localization sequence previously reported[23], while the role of the other (AAs 179–208) on EPAC1 nuclear localization was unknown. As shown in Supplementary Fig. 1b, mutants lacking each of these regions were slightly less expressed as compared to the wild type and displayed a significant decrease in their levels in the nuclear fraction, independently of the activation status. These preliminary experiments could suggest that both sequences may have a role in the nuclear localization of EPAC1, however, the presence of EPAC1 in the nuclear fraction pointed to the necessity to further investigate the subcellular location of these mutants. To better understand the role of each sequence, we deleted these regions in an EPAC1 construct that was tagged in its amino-terminus (N-terminus) with YFP (hereafter EPAC1-YFP)[30]. As shown in Supplementary Fig. 1c EPAC1-YFP was present in the nucleus and the nuclear envelope as expected[23,27]. On the other hand, the EPAC1^Δ732-764^-YFP mutant appeared soluble and did not enter the nucleus neither engaged with the nuclear envelope

(Supplementary Fig. 1d) as previously reported[23,27]. Interestingly, deletion of the residues 179–208 abolished the nuclear localization of EPAC1 without, however, affecting its ability to localize at the nuclear envelope (Supplementary Fig. 1e). The ability of the EPAC1^Δ179-208^-YFP mutant to engage with the nuclear envelope was not surprising since this mutant contains the residues reported to be important for the binding to RANBP2[23,27]. Taken together our data suggest that at least two aminoacidic domains are necessary for the entry of EPAC1 in the nucleus and point at the existence of a molecular mechanism that guarantees the presence of EPAC1 in this compartment.

The identification of specific sequences involved in the entry of EPAC1 in the nucleus is a strong indication for a functional role in this compartment. To test the responses of nuclear EPAC1 (nEPAC1) to cAMP, we overexpressed EPAC1-YFP in HEK cells, which are particularly fit as a model because they have been reported to be EPAC1-deficient[31,32], as also shown in Fig. 1a (non-transfected samples). As shown in Fig. 1b & Supplementary Movie 1, in untreated cells EPAC1-YFP localized in the cytosol, nuclear envelope, and nucleus. In response to intracellular cAMP elevation, elicited by challenging the cells with forskolin (FSK), a broad activator of transmembrane adenylyl cyclases combined to 3-isobutyl-1-methylxanthine (IBMX) to inhibit its degradation by phosphodiesterases, cytosolic EPAC1-YFP rapidly moved to the plasma membrane, as expected[7]. On the other hand, nuclear EPAC1-YFP oligomerized in well-defined round structures in approximately 40% of the EPAC1-YFP expressing cells (Fig. 1b, c). These structures did not depend on the fluorescent tag since both, a C-terminally mCherry2-tagged (mCherry2-EPAC1) and an untagged EPAC1 formed similar puncta in the nucleus of HEK cells in response to cAMP (Supplementary Fig. 2a, b). Importantly, the generation of nEPAC1 oligomers is independent of PKA activity since the PKA inhibitor H89 was unable to block their insurgence (Supplementary Fig. 2c second panel). In addition, the EPAC-specific cell permeant cAMP analog 8-pCPT-2'-O-Me-cAMP-AM (8CPT-cAMP) induced the formation of EPAC1-YFP puncta (Supplementary Fig. 2c third panel). Nuclear EPAC1 oligomers were also formed when cells were challenged with the G-Protein coupled receptor agonist norepinephrine (NE) which produced a much smaller cAMP increase (roughly 50% of FSK/IBMX), as measured by a FRET-based cAMP sensitive sensor[2] (Supplementary Fig. 2d). In response to cAMP binding, EPAC1 exits its autoinhibited state assumes an active conformation and activates the small GTPases Rap1 and Rap2[17,33]. To test whether Rap1&2 activation is important for nuclear oligomer formation we used a catalytically dead mutant (EPAC1^TF781-782AA^-YFP)[34] and found that was able to oligomerize in response to cAMP suggesting against the involvement of these proteins (Supplementary Fig. 2e). Thus, we conclude that upon cAMP elevation, EPAC1 forms nuclear puncta/oligomers.

Due to the structural similarities between EPAC1 and EPAC2, we tested whether the insurgence of nuclear structures was prerogative of EPAC1 or of both EPACs. As shown in Supplementary Fig. 3a, in cell fractionation experiments we found no evidence of EPAC2 localization in the nucleus. These experiments were in line with imaging experiments indicating that a construct of EPAC2 tagged with GFP did not enter the nucleus and remained soluble in cells challenged with cAMP-generating agonists (Supplementary Fig. 3b, c).

To test whether the formation of nuclear structures could be recapitulated by endogenous EPAC1, we used two different cell models, Human Umbilical Vein Endothelial Cells (HUVEC) and an ovary adenocarcinoma cell line (SKOV3) both expressing endogenous EPAC1 in the nucleus as confirmed by nuclear fractionation and Western Blotting (Fig. 1d). In cells treated with DMSO (vehicle control), endogenous EPAC1 appeared mostly soluble with a small number of oligomers. Nuclear EPAC1 oligomerization was drastically enhanced when intracellular cAMP levels were increased by challenging the cells for 30

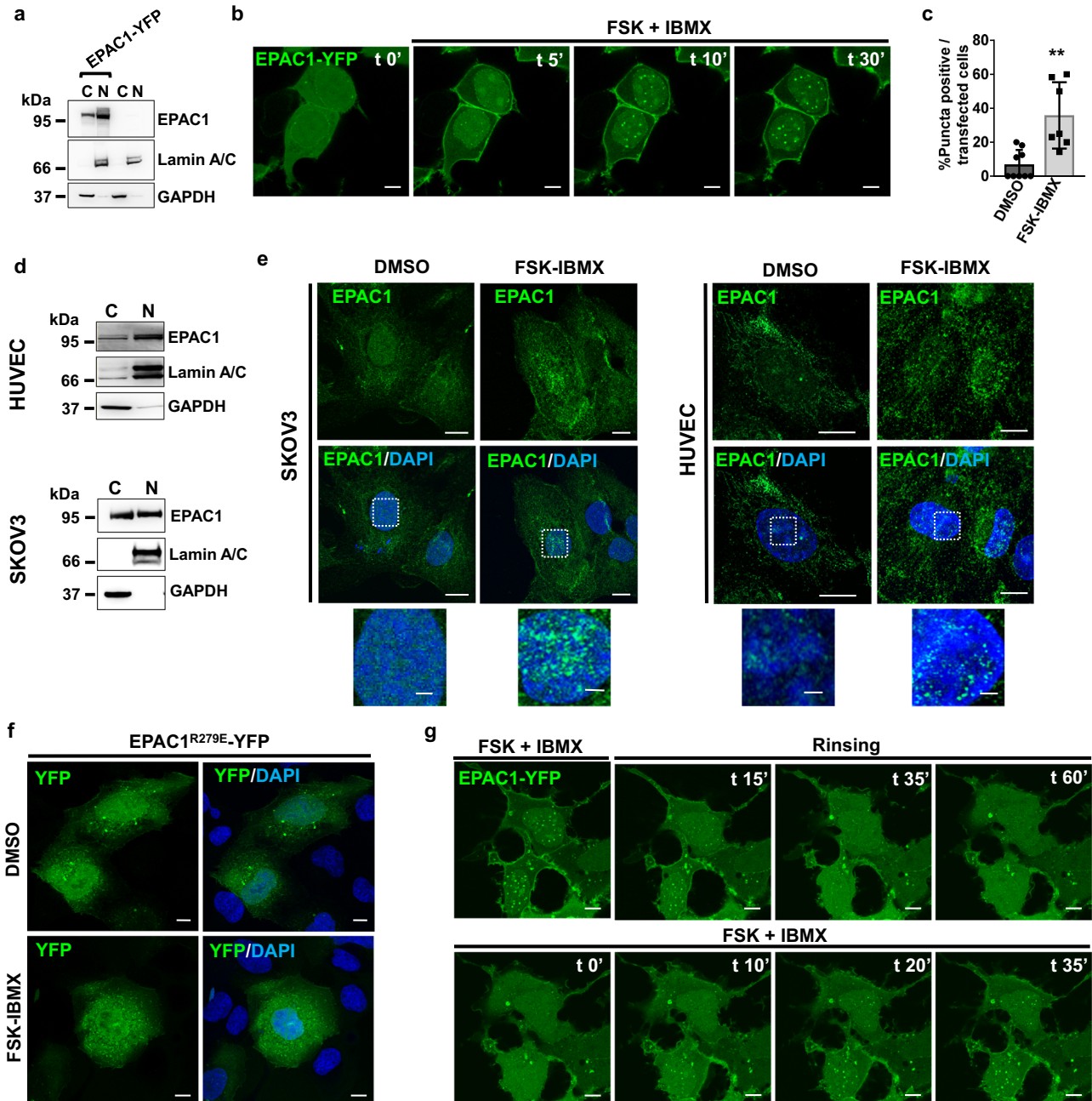

**Fig. 1 | The nuclear EPAC1 moiety forms reversible oligomers in response to cAMP elevations. a** Western blotting of cytosol and nuclei-enriched fractions of naïve (right lanes) or EPAC1-YFP expressing HEK cells (left lanes). **b** Still captures of live confocal imaging of EPAC1-YFP in HEK cells. In response to cAMP elevating agonists (forskolin (FSK) combined to IBMX cytosolic EPAC1 moves to plasma membrane while nuclear EPAC1 forms round-shaped structures. Scale bar, 5 µm. **c** Quantification of EPAC1-YFP expressing HEK cells forming nuclear EPAC1-YFP oligomers (over the total transfected cells) treated with FSK-IBMX for 1 hour. Data are expressed as mean ± SD and statistical significance was determined by two-sided, unpaired Student's t-test (**$p = 0.0013$). $n = 3$ independent experiments. Source data are provided as a Source data file. **d** Western blotting of cytosol and nuclei-enriched fractions of HUVEC and SKOV3 cells. **e** Confocal photomicrographs

of endogenous EPAC1 distribution in cells (HUVEC, SKOV3) treated with DMSO (control) or forskolin in combination to IBMX (FSK-IBMX) to increase cAMP levels. Scale bar, 10 µm (enlargements 3 µm). **f** Confocal photomicrographs of HEK cells expressing the cAMP-binding deficient mutant EPAC1[R279E]-YFP. Scale bar, 10 µm. **g** Live confocal imaging of EPAC1-YFP. Cells were pre-treated with FSK-IBMX to induce nuclear EPAC1-YFP oligomerization for 30 minutes. Upon rinsing EPAC1 oligomers dissolve to be formed again in response to subsequent treatment with FSK-IBMX. [FSK] 20 µM, [IBMX] 400 µM. Scale bar, 8 µm. Lamin A/C and GAPDH nuclear and cytosolic markers, respectively. Nuclei were visualized using DAPI. C: cytosol; N: nucleus. Experiments were repeated at least three times with similar results.

to 40 min with FSK combined to IBMX (Fig. 1e). We also noted that this treatment had different effects on the cytosolic EPAC1 which moved to plasma membrane or mitochondria depending on the cell type[12]. In line with these observations, a cAMP binding-deficient EPAC1 (EPAC1[R279E]-YFP)[30] was unable to form oligomers (Fig. 1f)

demonstrating that cAMP is necessary and sufficient to trigger nEPAC1 oligomerization. Collectively these data suggest that nEPAC1 oligomers may represent a different signaling modality through which cAMP signals are interpreted in the nucleus. However, to be considered a signaling event, nEPAC1 oligomers should be reversible and

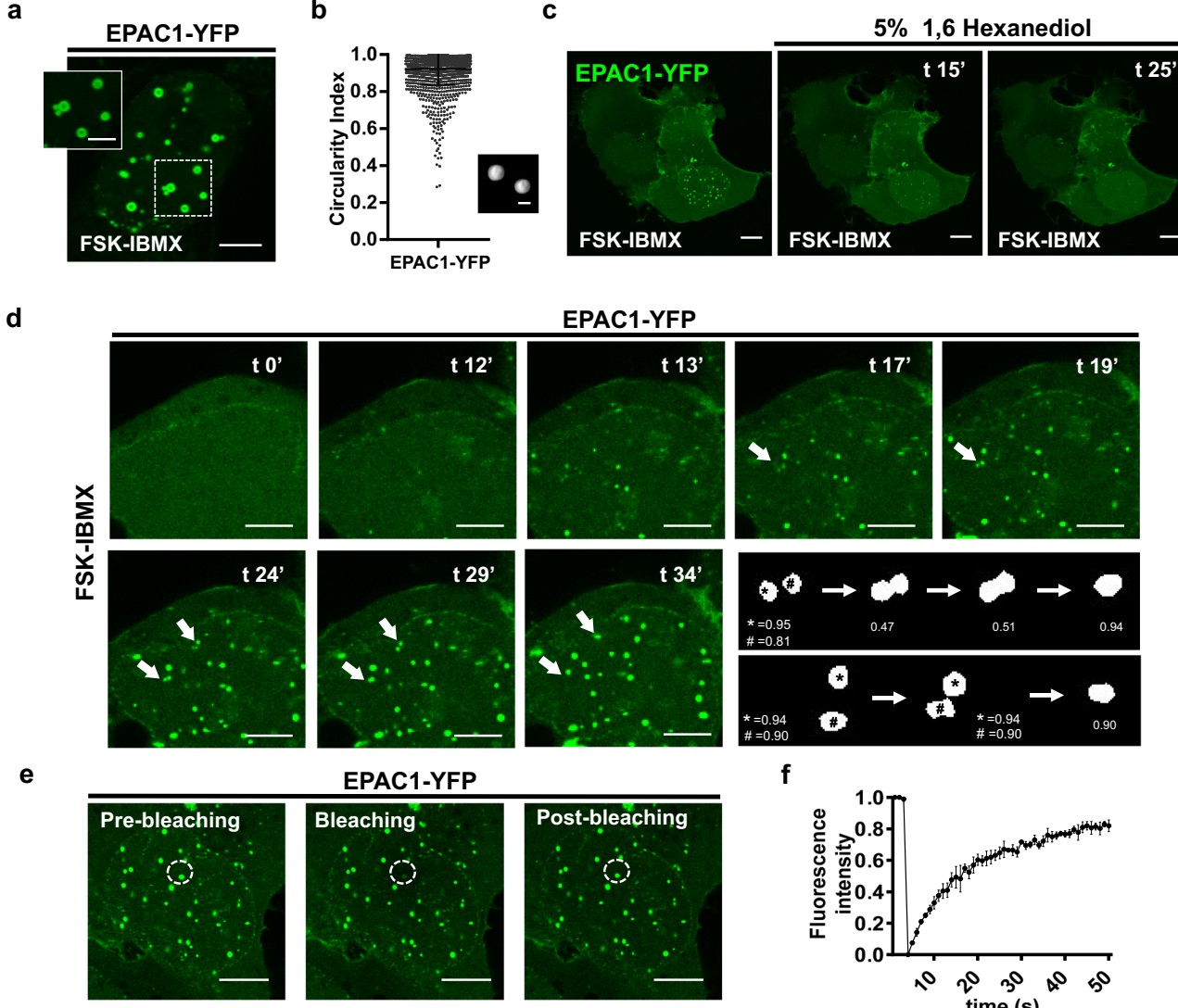

**Fig. 2 | Nuclear EPAC1 oligomers are highly dynamic structures with biomolecular condensate characteristics. a** Confocal photomicrograph showing the characteristic round hollow shape of EPAC1-YFP condensates after 60 min of FSK-IBMX treatment. Scale bar, 8 μm (enlargement: 2 μm). **b** Analysis of EPAC1 condensate circularity index and a representative condensate 3D rendering (inset), $n = 1595$ puncta, scale bar 0.8 μm. Data are presented as scatter dot plots of individual data points and mean ± SD error bars. **c** Still captures of live confocal imaging of HEK cells expressing EPAC1-YFP. Addition of 5% 1,6-hexanediol on top of FSK-IBMX disperses the nuclear EPAC1-YFP condensates. Scale bar, 10 μm. **d** Live confocal imaging of HEK cells expressing EPAC1-YFP. Treatment with FSK-IBMX triggers the formation of condensates and several fusion events of adjacent condensates are evidenced (arrows). Scale bar, 5 μm. In the last panel a rendering and calculation of the circularity index of two fusion events (*,#) is presented. **e** Still images of live confocal FRAP experiments (Fluorescence Recovery After Photobleaching). The fluorescence of a single nuclear EPAC1-YFP condensate was targeted with maximal laser power which bleached the fluorescent molecules present in the condensate. Fluorescence rapidly recovered demonstrating the exchange with unbleached molecules from the soluble surrounding. Scale bar, 10 μm. **f** Quantification of fluorescence intensity of four independent experiments shows 80% of recovery within 60 s from the bleaching event. [FSK] 20 μM, [IBMX] 400 μM. Data are presented as mean values ± SD. Source data are provided as a Source data file. Unless otherwise stated experiments were repeated at least three times with similar results.

reproducible in response to cAMP. To test this possibility, we treated cells expressing EPAC-YFP with cAMP elevating agonists which induced the formation of nuclear puncta. As shown in Fig. 1g and Supplementary Movie 2 EPAC1-YFP puncta rapidly dissipated after rinsing away the treatment and rapidly reformed in the next round of stimulation, substantially reproducing the behavior of a signaling event.

## Nuclear EPAC1 forms biomolecular condensates in response to cAMP

The dynamic nature of nEPAC1 oligomers suggested that these structures could be biomolecular condensates or membraneless organelles[35]. To verify the nature of nEPAC1-based oligomers we thus tested several indicators of condensation[36]. Biomolecular condensates tent to adopt spherical shape in order to minimize the surface area under the influence of interfacial tension. As shown in Fig. 2a, quantified in Fig. 2b and rendered in Supplementary Movie 3, in Airyscan super-resolution microscopy experiments nuclear EPAC1 puncta assumed a characteristic spherical shape[37,38]. Biomolecular condensates are thought to generate through weak, multivalent, and dynamic interactions among proteins and/or nucleic acids in the absence of a bounding membrane[35,39]. As shown in Fig. 2c & Supplementary Movie 4, nEPAC1-YFP oligomers were rapidly dissolved when 1,6-hexanediol, an aliphatic alcohol that interferes with weak

hydrophobic interactions[40] was added in the FSK-IBMX-complemented solution. While hexanediol sensitivity points to the importance of hydrophobic interactions in the maintenance of EPAC1 puncta is not diagnostic of condensation[36,41]. Nuclear EPAC1 oligomers could also undergo fusion events, another indicator suggesting that nEPAC1 condensates behave similarly to a viscus liquid phase[36], further substantiating the dynamic liquid-like nature of these structures (Fig. 2d & Supplementary Movie 5). As expected, while during the fusion process there is some loss of circularity, the resulting condensate rapidly assumes a spherical shape (Fig. 2d last panel). Finally, Fluorescence Recovery After Photobleaching (FRAP) experiments (Fig. 2e & Supplementary Movie 6) demonstrated that nEPAC1-YFP condensates rapidly recovered (quantified in Fig. 2f) after laser-induced bleaching, confirming the dynamic nature of these structures.

## The N-terminal region of EPAC1 is necessary for cAMP-dependent phase separation

The previous experiments suggested that nEPAC1 condensates could form through Phase Separation (PS), albeit with some EPAC1-specific features. For instance, while PS is strictly connected to the concentration of the transitioning molecule[42,43], nEPAC1 in its cAMP-free state does not form condensates independently of expression levels (Fig. 1b). In its inactive state, EPAC1 assumes an autoinhibitory conformation where the regulatory region sterically blocks its catalytic site. Cyclic AMP binding induces a conformational change that causes the regulatory lobe to move and expose the catalytic domain[12]. In this open conformation the nuclear moiety of EPAC1 can form condensates, pointing to the existence of domains that once exposed can participate or trigger the process of condensation. Intrinsically disordered regions (IDRs) do not adopt a fixed three-dimensional structure but instead exist in a heterogeneous collective of conformations[44] and can be enriched in proteins able to undergo PS[45]. As shown in Fig. 3a in silico analysis using the algorithm $(D^2p^2)$[46] identified several IDRs within EPAC1, especially within its catalytic (C-terminal) and regulatory (N-terminal) domains. When overlapped to the structure of EPAC1 (Fig. 3a) most of the predicted IDRs were found between folded domains, a topology that suggested roles more compatible to disordered linkers[47]. For this reason we focused on the N-terminal region and found that a deletion mutant of EPAC1 lacking the first 148 AAs (EPAC1$^{\Delta2-148}$-YFP), was unable to form condensates (Fig. 3b) independently of its expression levels as demonstrated by dose-response overexpression experiments transfecting 1-2-3 μg of construct (Supplementary Fig. 4a), and its activation in response to cAMP, even though it retains intact its ability to enter the nucleus, bind to cAMP and undergo conformational changes[34].

To test whether the N-terminal regulatory region is necessary for the formation of EPAC1 condensates in response to cAMP in the absence of other cellular components or factors, we tested the ability of purified EPAC1$^{WT}$ and EPAC1$^{\Delta2-148}$ proteins to form condensates in vitro. As shown in Fig. 3c (full traces) and Supplementary Fig. 4b–d in the absence of cAMP, only a minimal fraction of purified EPAC1$^{WT}$ condensed in response to increasing salt concentrations. On the other hand, upon addition of cAMP, a significantly larger portion of EPAC1$^{WT}$ switched to condensates. In line with its behavior in living cells, EPAC1$^{\Delta2-148}$ was unable to form condensates independently of salt concentrations and cAMP levels (Fig. 3c dotted traces). Of note, EPAC1$^{WT}$ reached its maximum condensation ratio (+cAMP/-cAMP = ~2.5) at a near-physiological concentration of salt (150 mM) (quantified Fig. 3d). To better define the role of the regulatory region of nEPAC1 in condensate formation we decided to generate several other mutants by deleting shorter fractions of the N-terminal region. Interestingly, deletion of AAs 2–24 (EPAC1$^{\Delta2-24}$-YFP) abolished the ability of nEPAC1 to generate condensates (Fig. 3e) as did the deletion of a region containing the DEP domain (AAs 48–148) EPAC1$^{\Delta48-148}$-YFP (Fig. 3f). We also

noted that the plasma membrane localization of these mutants in response to cAMP was impeded, however, this was expected since the N-terminal region is responsible for EPAC1 membrane localization[12]. On the contrary, the ability of mutants lacking intermediate AAs, EPAC1$^{\Delta25-50}$-YFP and EPAC1$^{\Delta51-73}$-YFP to generate condensates remained unaffected as shown in Supplementary Fig. 4e and f, respectively. Multivalent IDRs are thought to facilitate protein interactions or recruitment of proteins to condensates[48]. We reasoned that the failure of EPAC1$^{\Delta2-148}$ EPAC1$^{\Delta2-24}$ and EPAC1$^{\Delta48-148}$ to form puncta, could be explained either by their inability to initiate EPAC1 clustering or by perturbed recruitment to the forming condensates. To test this hypothesis, we co-expressed each mutant together with EPAC1$^{WT}$ or mCherry-EPAC1$^{WT}$, both able to condensate. To ensure similar expression of the two constructs and avoid artifacts, we performed the analysis 18–24 h after transfection. As shown in Supplementary Fig. 5a, b, both EPAC1$^{\Delta2-24}$ and EPAC1$^{\Delta48-148}$ were able to participate to the forming condensates of an untagged EPAC1$^{WT}$. On the other hand, EPAC1$^{\Delta2-148}$ did not participate in the condensates formed by the untagged EPAC$^{WT}$ (Supplementary Fig. 5c). We reasoned that the behavior of EPAC1$^{\Delta2-148}$ could depend ether by its inability to participate in the formation or could be underlined by a dominant negative effect of the mutant on the co-expressed condensate-capable construct. To test this possibility, we co-expressed EPAC1$^{\Delta2-148}$-YFP together with EPAC1-YFP and as shown in Supplementary Fig. 5d, several condensates were evidence in response to cAMP indicating that EPAC1-YFP retained its ability to condense in the presence of EPAC1$^{\Delta2-148}$-YFP. Together, these data suggest that both N-terminal regions (AAs 2–24 and 48–148) are necessary for the insurgence of EPAC1 condensates, while each of them is sufficient to drive the recruitment of EPAC1 to its condensates.

In addition to the two domains (AAs 2–24 and 48–148), we also identified an aminoacidic region (AAs 145–175) that appears to be crucial for the cAMP-dependent regulation of EPAC1 condensation. In fact, as shown in Fig. 3g, a deletion mutant lacking this region (EPAC1$^{\Delta145-175}$-YFP) constitutively forms condensates independently of the presence of cAMP. Taken together, these data indicate that EPAC1 can form condensates in the nucleus thanks to the combination of specific regions within its N-terminal region and conformational changes triggered by cAMP that likely increase the multivalency of the protein[44].

## Nuclear EPAC1 condensates interact with other nuclear membraneless organelles

In recent years nuclear membraneless organelles emerged as central regulators of a plethora of nuclear processes, from transcription to RNA processing to chromosome structure and maintainance[35]. Nuclear condensates can stem thanks to the clustering of a single protein, however, there are multimolecular structures where several proteins and other macromolecules participate functionally and structurally[35,49,50]. The discovery that EPAC1 is a condensation-proficient protein, raised the question of whether it participated or interacted with already characterized nuclear membraneless organelles. As shown in Fig. 4a, b, nEPAC1-YFP structures did not interact with nucleoli, while only sartorially engaged with cajal bodies as demonstrated by immunofluorescence using their respective markers nucleolin[51] and Survival Motor Neuron protein (SMN)[52]. On the contrary, in several experiments was clear that nEPAC1 condensates significantly overlapped with pro-myelocytic leukemia protein (PML)-based nuclear bodies (PML-NBs) and the Nuclear Protein of the ATM Locus (NPAT), a marker of Histone Locus Bodies (HLBs)[52], Figs. 4c and 4d, respectively. These interactions were quantifiable by calculating the degree of overlap, or the distance between the centroids of nEPAC1-YFP condensates to their neighboring structures (Fig. 4e). Interestingly, 3D reconstruction of Airyscan super-resolution images suggested different

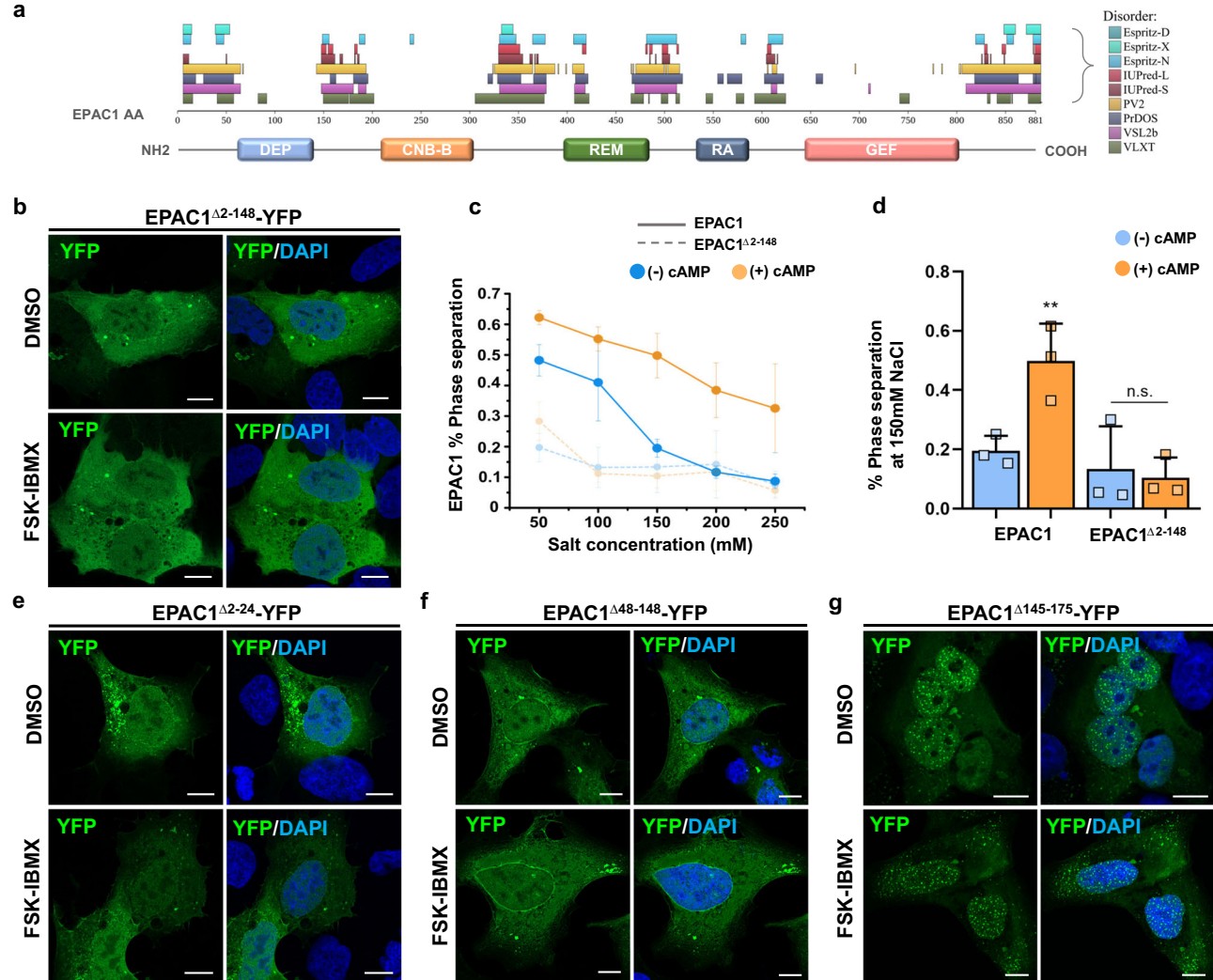

**Fig. 3 | The N-terminal region of EPAC1 contains low complexity regions necessary for phase transition. a** Schematic map of the EPAC1 domains overlapped to the distribution of intrinsically disordered regions within EPAC1 identified by the bioinformatic algorithm $D^2p^2$, which takes advantage of several specialized IDR prediction tools (listed on the right). **b** Confocal images of HEK cells expressing the deletion mutant EPAC1$^{\Delta2-148}$-YFP. Scale bar, 10 μm. **c** Proportion of purified wild type EPAC1 that forms condensates in function of the salt concentration and the presence or absence of cAMP. The difference is maximal at salt concentrations comparable to those of physiological conditions (150 mM), indicating that when cAMP is not present, the amount of EPAC1 able to form condensate is minimal, while it reaches a value of ~50% when cAMP is available, in good agreement with the cell biology experiments. Data are presented as mean

values ± SEM. $n = 3$ independent experiments. **d** Percentage of phase separation at near-physiological conditions of purified EPAC1 compared with the EPAC1$^{\Delta2-148}$ mutant, not able to form condensates physiologically. Data are presented as mean values ± SEM. $n = 3$ independent experiments. Statistical significance was determined by One-way ANOVA (**$p = 0.01818$). **e** Confocal images of HEK cells expressing the deletion mutant EPAC1$^{\Delta2-24}$-YFP or EPAC1$^{\Delta48-148}$-YFP (**f**). Both mutants were unable to undergo phase transition in response to FSK-IBMX treatment. **g** Confocal images of HEK cells expressing the deletion mutant EPAC1$^{\Delta145-175}$-YFP. This construct formed condensates constitutively and independently of the cAMP levels. Nuclei were visualized using DAPI. [FSK] 20 μM, [IBMX] 400 μM. Scale bar 10 μm. Experiments were repeated at least three times with similar results.

modalities of interaction between nEPAC1 condensates and these membraneless organelles. As shown in Fig. 4f cajal bodies (SMN) and HLBs (NPAT) seem to engage with EPAC1 condensates without complete fusion, on the other hand PML-NBs interact more extensively. Taken together, these data suggested that nEPAC1 could be recruited and participate in condensates with other condensate-proficient proteins. However, we argue that EPAC1 is not a crucial structural component of cajal bodies, HLBs and PML-NBs since all of these condensates were present in unstimulated HEK cells where nuclear EPAC1-YFP is diffused throughout the nucleoplasm (DMSO treatments). These considerations raise the possibility that nEPAC1 could be recruited in response to cAMP and may exert a functional or a regulatory role on other nuclear bodies.

## Nuclear EPAC1 condensates regulate histone transcription in a cAMP-dependent manner

A primary function of nuclear membraneless organelles is to regulate transcription. For instance, HLBs contain factors required for processing histone pre-mRNAs[52], while PML-NBs have been found to associate with transcriptionally active sites[53]. To test the involvement of nEPAC1 condensates in transcriptional regulation we used high throughput whole transcriptome RNA-sequencing. In our experimental design, illustrated in Fig. 5a, EPAC1-deficient HEK cells[31] (Fig. 1a) were transfected with EPAC1-YFP or, as control, the EPAC1$^{\Delta2-148}$-YFP mutant which is unable to phase separate (Fig. 3b, c). Twenty-four hours after transfection, cells were treated with the cell permeant EPAC-selective cAMP analog 8CPT-cAMP (5 μM) to induce

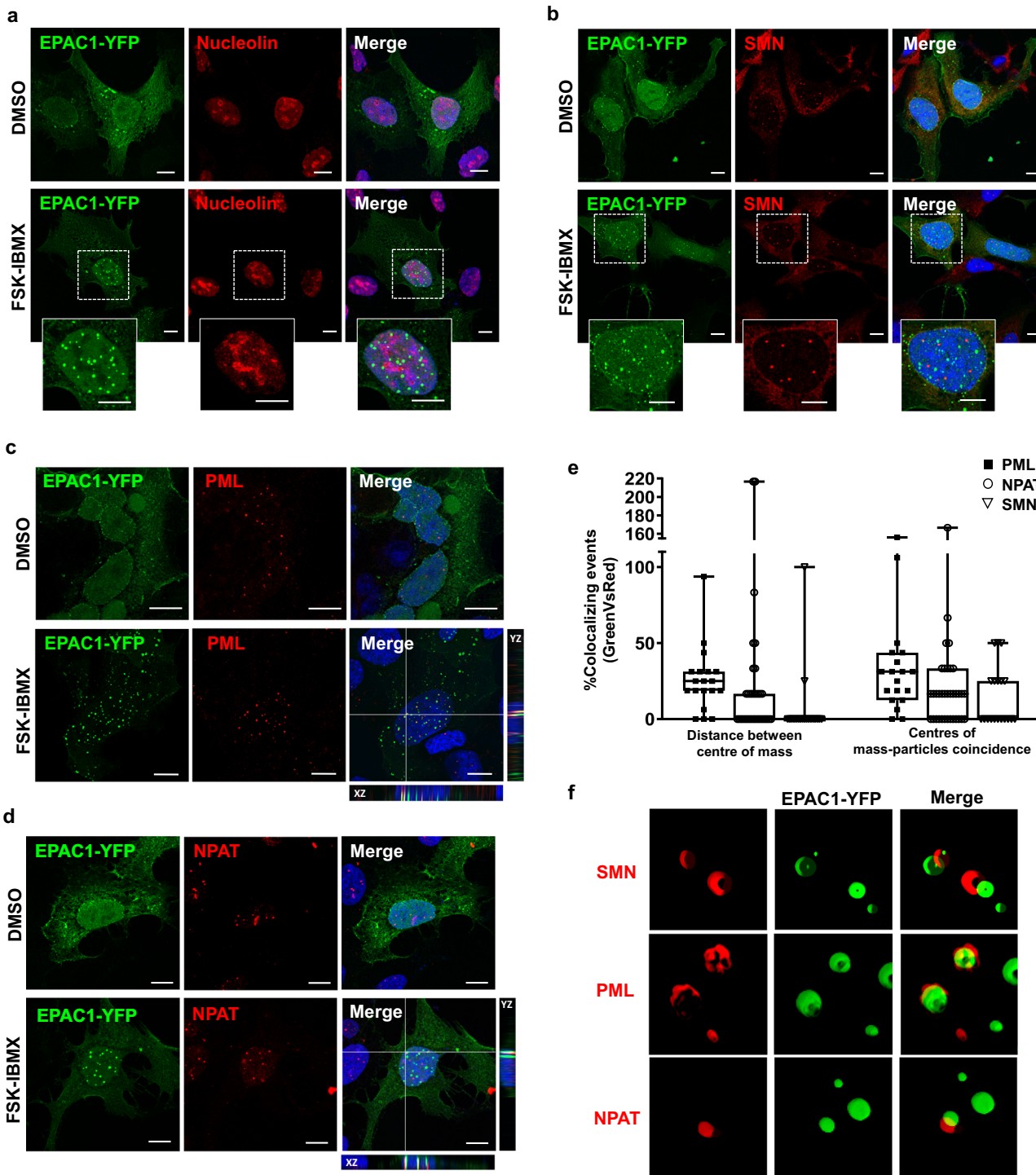

**Fig. 4 | EPAC1 condensates interact with other nuclear membraneless organelles.** Confocal photomicrographs of HEK cells expressing EPAC1-YFP and probed for endogenous Nucleolin for identifying the nucleoli (**a**) or SMN (survival motor neuron) protein to identify Cajal bodies (**b**). **c** Confocal images of endogenous PML (red) to map PML-NBs and EPAC1-YFP (green). Several overlapping spots were observed between the two organelles (orthogonal view in last panel). **d** Confocal images of endogenous NPAT (Nuclear Protein, Ataxia-Telangiectasia Locus) (red) to recognize histone locus bodies (HLBs) and EPAC1-YFP (green). Several points of overlap were observed (orthogonal view in last panel). Nuclei were visualized using DAPI. [FSK] 20 µM, [IBMX] 400 µM. Scale bars, 10 µm. Experiments were repeated

at least three times with similar results. **e** Colocalization analysis based on center of mass-particles coincidence or based on distance between centers of mass (green Vs red) between EPAC1-YFP and PML, NPAT and SMN. The boxes show interquartile ranges. The horizontal line across each box denotes the median, and vertical lines extending above and below each box indicate the minimum and maximum values. PML *n* = 17, NPAT *n* = 35, SMN *n* = 19 nuclei over 3 independent experiments. **f** 3D representation of image volume rendering (Airyscan confocal 3D volume rendering of z-stack images). The experiments for (**a**), (**b**), (**c**), (**d**) were repeated 3–5 times independently with similar results. Source data are provided as a Source data file.

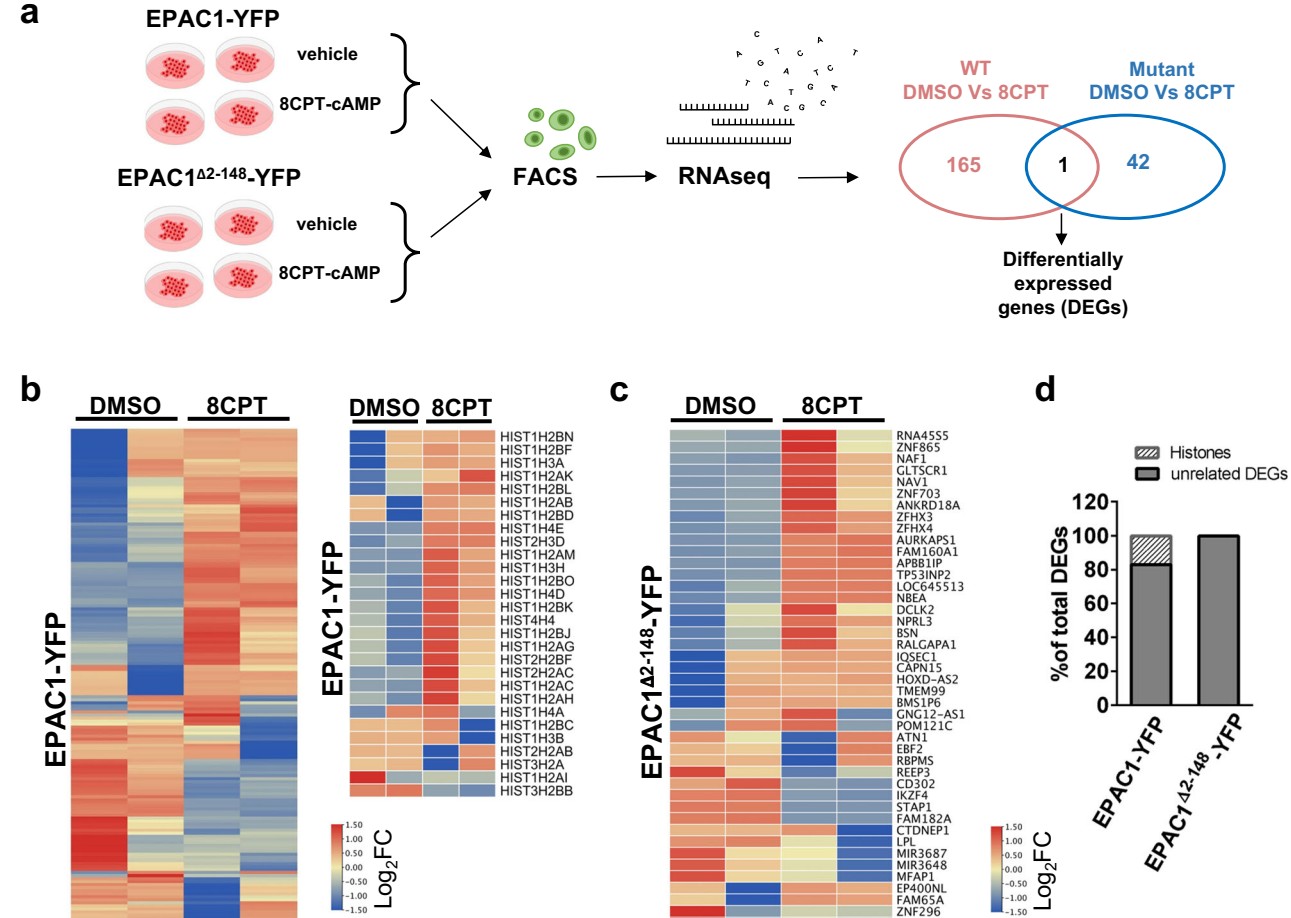

**Fig. 5 | EPAC1 condensates regulate the transcription of several genes including the large Histone cluster 1. a** Schematic representation of the experimental design used for the RNAseq experiments (created using Microsoft PowerPoint). **b** Heatmap of differentially expressed genes (DEGs) between EPAC1-YFP-expressing HEK cells treated with DMSO or the EPAC1-specific activator 8CPT-cAMP (5 μM) for 40′ ($p < 0,01$). Significance was calculated using Cuffdiff's t-test analogical methods[56]. Upregulated genes are represented in red while downregulated ones are in blue. Detail: histone genes differentially expressed. **c** Heatmap of DEGs between EPAC1$^{Δ2-148}$-YFP expressing HEK cells treated with DMSO or 8CPT-cAMP (5 μM). **d** Percentage of differentially expressed histone genes in the two comparisons. RNAseq experiments were repeated three times independently, for analysis two experiments were used for EPAC1-YFP and three for EPAC1$^{Δ2-148}$-YFP. Source data are provided (see "Data availability" section).

condensate formation, or with DMSO (vehicle) as control for 40 min. We strategically choose 8CPT-cAMP because of its inability to activate PKA and its high affinity for EPAC1 as compared to EPAC2[54]. On the other hand, the 40 min timepoint was chosen based on the reports identifying the cAMP-dependent transcription peak between 30 and 60 min[55]. After treatment, EPAC1-expressing cells were selected by Fluorescence-Activated Cell Sorting (FACS) using the YFP fluorescence and total RNA was extracted. Directional RNA sequencing of rRNA-depleted total RNA generated an average of ~76 million reads per sample, of which 64% to 75% could be aligned to the reference genome, suggesting very good coverage and sequencing depth with low ribosomal RNA contamination. RNA-seq data analysis identified 21,705 annotated genes as expressed in at least one of the sequenced samples. Principal Component Analysis (PCA) indicated anomalous behavior for one of the three biological replicates of the EPAC1-YFP treated with 8CPT-cAMP, which was not included to further analyses. We used CufDif2[56] for the identification of differentially expressed genes in all samples ($p$-value ≤ 0.01). When compared to untransfected cells, overexpression of EPAC1-YFP affected the transcription of 662 genes while overexpression of EPAC1$^{Δ2-148}$-YFP impinged on the expression of 1803 genes (Supplementary Fig. 6a, b). Importantly, in non-transfected cells EPAC1 transcripts were barely detectable, which was in line with the absence of EPAC1 in Western Blotting experiments (Fig. 1a), while

the expression levels of the two constructs in the transfected populations were virtually identical as indicated by the fold increase compared to the untransfected cells calculated in the RNAseq experiments (log$_2$Fold 12.89 for EPAC1-YFP and 12.74 for EPAC1$^{Δ2-148}$-YFP). As shown in Fig. 5b, 40 min treatment of EPAC1-YFP expressing cells with 8CPT-cAMP, induced modest but significant changes in the expression of 166 genes as compared to untreated cells expressing the same construct. On the other hand, when EPAC1$^{Δ2-148}$-YFP was expressed, the same treatment affected the expression of only 42 genes as compared to untreated cells (Fig. 5c). The effect of 8CPT-cAMP in the transcriptional signature of untransfected HEK cells was negligible (Supplementary Fig. 6c), further confirming the absence of EPAC1 in HEK cells and, most importantly, suggesting that the differences between the effects of the two constructs depended exclusively on their ability to form or not nuclear condensates. When we compared the two sets of genes regulated by 8CPT-cAMP treatment in EPAC1-YFP and EPAC1$^{Δ2-148}$-YFP we found no significant overlap (only one gene was in common). Interestingly, further analysis revealed that upon activation, EPAC1-YFP but not EPAC1$^{Δ2-148}$-YFP, affected the transcription of 77 nuclear proteins, among which 28 (36%) were histones (Fig. 5b right panel). Most of these histone genes (20/28) (71%) located to the large cluster of histone genes on human chromosome 6 (6p21–p22), representing 16.9% of the differentially expressed genes (DEGs)

(Fig. 5d). In line with this observation, histone genes were differentially expressed only in response to EPAC1-YFP overexpression (13/662 DEGs) as compared to naive HEK cells. While, on the other hand, overexpression of the EPAC1$^{\Delta2-148}$-YFP had a negligible effect on histone expression (3/1803 DEGs). The strikingly high incidence of histones in our RNAseq analysis together with our previous observation that EPAC1-YFP condensates colocalized with the HLBs-marker NPAT (Fig. 4d), strongly suggested that the activation of the nuclear moiety of EPAC1 is a regulatory event of HLBs activity. Histones are organized in gene clusters and their transcription depends on several factors that are enriched in HLBs[52]. In mammalian cells, the synthesis of histone proteins is cell cycle-dependent and peaks during the S phase. This process is coordinated by the HLBs that act as platforms and recruit a set of specific transcription and pre-mRNA processing factors to the histone transcription loci[57]. Since EPAC1 was shown to be affected by and affect cell cycle progression[30] in a cell type-dependent manner[58], we questioned whether the effects observed on histone expression could be explained by changes in cell cycle due to EPAC1 activation, rather than being directly connected to the formation of nEPAC1 condensates. To exclude this possibility, we synchronized HEK cells expressing EPAC1-YFP using double Thymidine block and followed their cell cycle progression 1 and 8 h after treatment with 8CPT-cAMP for 40'–60' (like our RNAseq experiments). As shown in Supplementary Fig. 6d, cell cycle progression was indistinguishable between cells expressing EPAC1-YFP and treated ether with 8CPT-cAMP or DMSO, confirming that the effects of EPAC1 activation on histone transcription are independent of cell cycle progression variations.

While this manuscript was in revision, it was reported that EPAC1 condensates participate in the regulation of SUMOylation, a process that entails the addition of small ubiquitin-related modifier (SUMO) to the target proteins[59]. Histone SUMOylation has been associated to the regulation of transcription, particularly its repression[60–62]. Although our data connect nEPAC1 condensates to increased histone expression, we tested the possibility that the transcriptional effects of 8CPT-cAMP could be linked to changes in the SUMOylation status of the cells. Given that HLBs and nEPAC1 condensates partially overlapped (Fig. 4d, e), we reasoned that 8CPT-cAMP treatment could increase the SUMOylation of HLB components thanks to their interaction with nEPAC1 condensates. To test this possibility, we performed immunofluorescence experiments where we simultaneously labeled HLBs (NPAT) and SUMO2/3 in cells expressing EPAC1-YFP treated with vehicle (DMSO) or 8CPT-cAMP. As shown in Supplementary Fig. 7a, b, we were able to identify overlapping foci of nEPAC1 with SUMO2/3 and NPAT however, we were unable to detect structures with all three proteins independently of the condition. These data suggested that the interaction of HLBs with nEPAC1 condensates does not affect the SUMOylation status of the former.

We next tested the functional significance of SUMOylation on histone expression induced by nEPAC1 condensates. As shown in Supplementary Fig. 7c and quantified in Supplementary Fig. 7d, treatment of HEK cells expressing EPAC1-YFP with 8CPT-cAMP induced a modest but significant increase in total SUMOylation with respect to untreated cells, detected by Western Blotting. While treatment with the SUMOylation inhibitor 2-(2,3,4-Trihydroxyphenyl)-4H-chromen-4-one (2D-08) completely reversed this effect (Supplementary Fig. 7c, d). To evaluate the effect of 2D-08 on the transcription of histones induced by 8CPT-cAMP we performed real-time PCR experiments using a probe for one of the regulated histone genes (Hist1h2ai). As expected, in HEK cells expressing EPAC1-YFP treatment with 8CPT-cAMP induced the expression of Hist1h2ai, and this effect persisted in the presence of the SUMOylation inhibitor 2D-08 (30 μM) (Supplementary Fig. 7e). These data clearly indicate that

nEPAC1 condensates regulate histone transcription independently of SUMOylation.

### Nuclear EPAC1 condensates localize at the histone cluster 1 locus

To determine whether nEPAC1 was directly involved in the regulation of histone transcription we designed a custom fluorescent probe for the large histone cluster 1 on chromosome 6p22.2 (covered region: chr6:26019341-26201862) and performed fluorescence colocalization experiments between EPAC1-YFP condensates and the Chromosome 6p22.2 region visualized by fluorescence in situ hybridization (FISH), in cells treated with FSK-IBMX (to increase cAMP levels) for 40 min. As shown in Fig. 6a, EPAC1-YFP-based condensates displayed a high degree of colocalization with the Chr6p22.2 probe, while no colocalization was observed between nEPAC1 condensates and three unrelated regions: the long arm of chromosome 21 (21q22.13-q22.2) (no EPAC1-dependent DEGs present) (Fig. 6b), the centromeric region of chromosome 15 (CEP15) (Fig. 6c), and the centromeric region of chromosome 12 (CEP12) (Fig. 6d). We next used a colocalization colormap method to generate an index of correlation (Icorr), which represents the fraction of positively correlated (colocalized) pixels and allows for a very sensitive quantitative measurement of colocalization. As shown in Fig. 6e the probe of Chr6p22.2 displayed the highest Icorr value, confirming the colocalization of this genomic locus and nEPAC1 condensates. On the contrary the Icorr values of the other three Chromosomic regions tested were low and indicated no colocalization with EPAC1-YFP (Fig. 6e right panel). Taken together our RNA-seq and fluorescence colocalization FISH experiments demonstrate that in response to cAMP elevations, the nuclear moiety of EPAC1 generates condensates in the proximity of the histone cluster 1 locus to regulate its transcription.

## Discussion

Nuclear membraneless organelles offer the appropriate three-dimensional organization and components necessary for guaranteeing precise and topologically restricted nuclear functions[35]. These structures gem in response to specific stimuli thanks to highly dynamic processes, however, the mechanisms through which the triggering signal is coupled to the condensate formation remain elusive. Here, we find that phase separation of nuclear EPAC1 can be considered a bona fide signaling event, controlled by the levels of the second messenger cAMP and impinging on transcription, and possibly on the function of nuclear membraneless organelles. In fact, nEPAC1 condensates were reported to regulate SUMOylation while in the present manuscript we find that cAMP-triggered nEPAC1 condensates colocalize with NPAT, a marker of histone locus bodies[57] and regulate the expression of a specific histone gene cluster at chromosome 6.

SUMOylation is a conserved posttranslational modification enriched in the nucleus where more than 70% of proteins are SUMO substrates[63]. Interestingly, proteins that contain IDRs tend to be prone to SUMOylation and consequently several nuclear condensates are sites of enriched SUMOylation[63,64]. This process is crucial for nuclear function and can regulate several processes including mRNA processing and metabolism[65], chromosome biology, and chromatin organization[63,66]. SUMOylation is also involved in the regulation of gene transcription, albeit mostly as an inhibitory event[62,63]. The finding that nEPAC1 condensates can regulate SUMOylation raised the possibility that this process could mechanistically explain the effects of nEPAC1 on histone expression. However, we find that the effects of nEPAC1 condensates on transcription are completely independent of SUMOylation. Based on these observations it is reasonable to hypothesize that the two functional outcomes of nEPAC1 condensates are decoupled and likely regulated by distinct mechanisms. One possibility could be

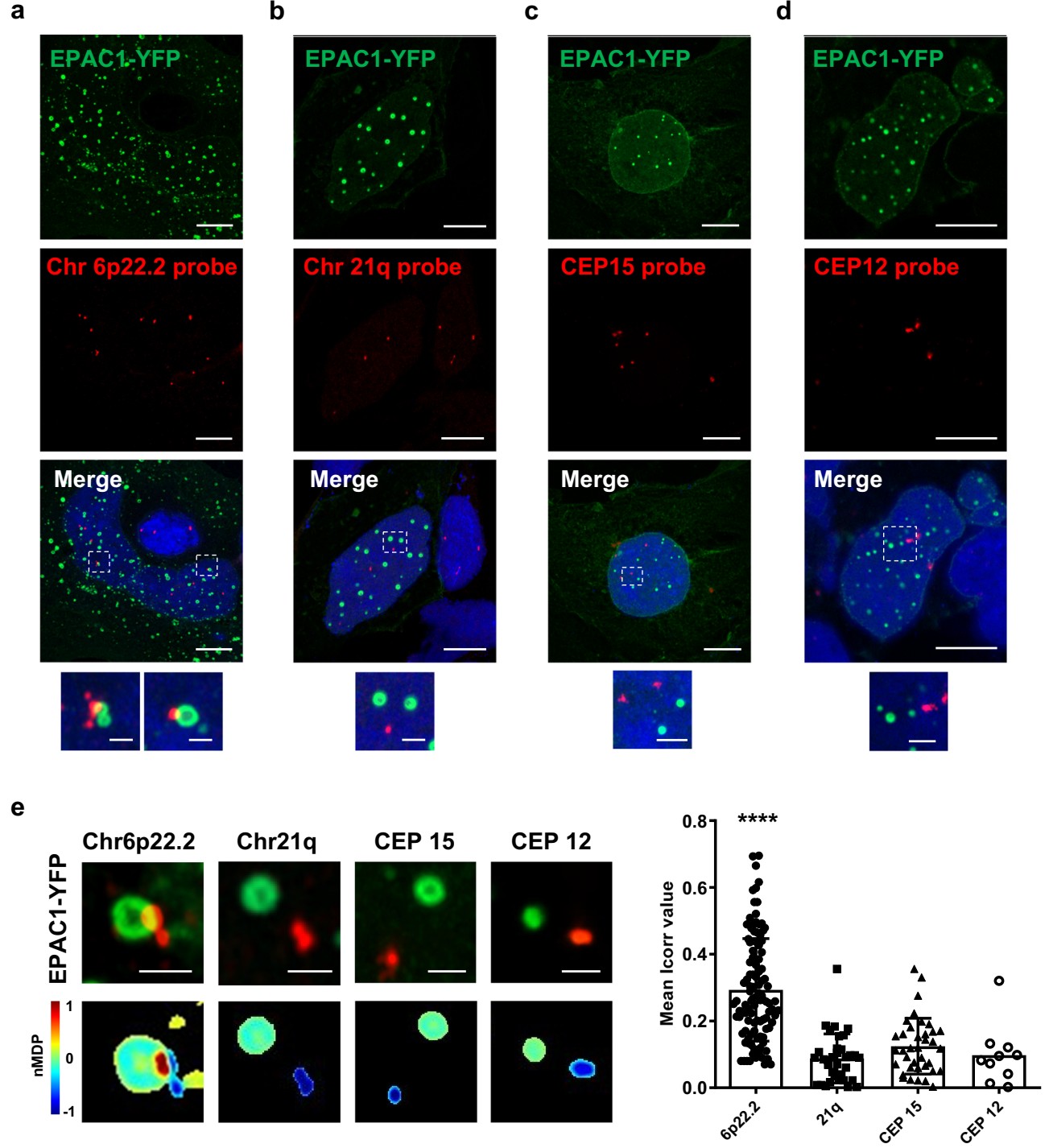

**Fig. 6 | EPAC1 condensates localize at the Histone cluster 1 on chromosome 6.**
**a** Confocal photomicrographs of colocalization experiments between EPAC1-YFP fluorescence (green) and the chromosome 6 p22.2 locus labeled by a custom-made probe (red) using fluorescence in situ hybridization (FISH) in HEK cells treated with FSK-IBMX. Significant overlap between the DNA probe and EPAC1-YFP was observed in around 40% of the labeled Chr 6p22.2 loci. **b**, **c**, **d** Confocal photo-micrographs of EPAC1-YFP (green) and, respectively, (red) chromosome 21 (21q22.13-q22.2), chromosome 12 centromere (CEP) and Chromosome 15 cen-tromere (CEP) loci labeled in FISH experiments using commercial probes, in HEK cells treated with FSK-IBMX. No significant correlation between these sites and

EPAC1-YFP (green) was observed. Scale bar, 10 μm (enlargement 1 μm).
**e** Correlation analysis between red and green signals. Lower panel shows colori-metric maps using nMDP values. (Right panel) Graph depicting the calculated index of correlation (Icorr) which represents the fraction of positively correlated (colo-calized) pixels (FISH foci analyzed Chr6p22.2 $n = 110$ over 4 independent experi-ments; Chr21q $n = 34$ over 3 independent experiments; CEP15 $n = 37$ over 3 independent experiments; CEP12 $n = 10$ over 2 independent experiments). Statis-tical significance was determined by one-way ANOVA (****$p < 0.0001$). Error bars: SD. Source data are provided as a Source data file. Nuclei were labeled with DAPI. [FSK] 20 μM, [IBMX] 400 μM. Scale bar, 1 μm.

that nEPAC1 condensates interact with, and impinge on, the function of other nuclear membraneless organelles. Indeed, we and others find that nEPAC1 interacts with PML[59] a protein well known to be SUMOylated[67], while in the present manuscript we demonstrate that nEPAC1 condensates are found in the proximity of and can overlap with HLBs, the membraneless organelles responsible for the cell cycle-dependent transcription of histones[68]. To control transcription, nEPAC1-condensates could provide a controlled modality for localizing with high efficiency to the active site important factors in a spatial and temporal-regulated manner. Histones are crucial chromatin components that are expressed during the DNA replication phase (S phase) as non-polyadenylated mRNAs from clusters located to specific chromosomic loci[57,68]. The coordination of histone transcription relies on the formation of histone locus bodies that contain components necessary for both the transcription and maturation processes[57]. Our data suggest that nEPAC1 condensates can engage the HLBs and likely increase their transcriptional activity, adding an additional regulatory element, the cAMP signaling cascade, to this process. Considering the importance of histone levels to cellular proliferation, the nuclear cAMP/EPAC1-condensate axis represents a molecular mechanism that could impinge on physiological or pathological cell division. Typically, increases in cAMP levels result in the increased enzymatic activities of its effector proteins. Our findings add a new level of complexity to this cascade as they show that in the case of the nuclear moiety of EPAC1, cAMP elevations result in the generation of hollow compartments that offer a privileged, and transient space where distinct components may be stored, or specific reactions may occur. Based on our findings, it is tempting to speculate that the nuclear moiety of EPAC1 provides the means through which cAMP controls the spatial compartmentalization of the nucleus.

## Methods

### Reagents
Forskolin (FSK), H-89 dihydrocloride (H89), 3-isobutyl-1-methylxantine (IBMX), dimethyl sulfoxide (DMSO), 1,6-Hexanediol, Phosphate Buffered Saline (PBS), Tween 20, Bovine Serum Albumin (BSA) and Skim Milk Powder were from (Merck KGaA, Darmstadt, Germany). 8-(4-Chlorophenylthio)-2′-O-methyladenosine-3′,5′-cyclic monophosphate, acetoxymethyl ester (8-pCPT-2′-O-Me-cAMP-AM) was from BioLog (Biolog Life Science Institute GmbH & Co. KG, Bremen, Germany). SUMOylation Inhibitor III, 2-D08 was from Calbiochem. Adenosine 3′,5′-cyclic monophosphate (cAMP) for in vitro experiments was from Sigma-Aldrich (A6885).

### Cell culture and transfection
HEK293 cells were purchased from ATCC (CRL-1573) and grown in Dulbecco's modified Eagle's high glucose medium (DMEM, ECM0728 Euroclone, Milan, Italy), supplemented with 1% penicillin–streptomycin (Life Technologies, 15140163), and 10% Fetal Bovine Serum (FBS) (Euroclone, Milan, Italy ECS0180). Ovarian adenocarcinoma SK-OV-3 cells were purchased from ATCC (HTB-77) and grown in RPMI (Sigma-Aldrich R8758) supplemented with 1% penicillin–streptomycin (Life Technologies, 15140163), and 10% Fetal Bovine Serum (Euroclone, Milan, Italy ECS0180L). Human Umbilical Vein Endothelial Cells (HUVEC) were a gift from the laboratory of Prof. Luca Scorrano (University of Padua). All cell lines were grown in a humidified incubator at 37 °C and 5% CO$_2$ atmosphere and tested for mycoplasma contamination every three months. Cells were split when 80–90% confluence was reached, every 2-3 days. For confocal imaging, cells were grown on glass coverslips, coated with poly-L-Lysine (Sigma-Aldrich, P4707). Twenty-four hours after plating cells were transfected with Lipofectamine 2000 Reagent (Thermo Fisher Scientific, Waltham, MA, USA) according to the manufacturer's instructions.

### In silico nuclear localization sequence (NLS) and disordered region enrichment analysis
The EPAC1 aminoacidic sequence was submitted in the NLS Mapper algorithm[29]. Two putative NLS sequences were identified and further analyzed by mutagenesis. The algorithm D2P2[46] was used to assay disorder propensity for each amino acid of EPAC1. Putative Intrinsically Disorder Regions (IDRs) were scored positively by multiple algorithms employed by the D2P2. IDRs that were within functional regions of EPAC1 (cAMP binding domain or Catalytic domain) were not further analyzed. IDRs at the amino terminus were further pursued by mutagenesis.

### Plasmids and mutagenesis
pEGFP-N3-EPAC1 and pEGFP-N3-EPAC1D2-148 were a gift from Prof. Xiaodong Cheng (University of Texas Health Science Center at Houston). EPAC1 subcloning in the vector mCherry2-C1, as well as EPAC1 mutagenesis, were performed using the Takara In-Fusion® HD Cloning kit (638910) according to the manufacturer's instructions. mCherry2-C1 was a gift from Michael Davidson (Addgene plasmid #54563; https://www.addgene.org/54563/). The sequences of the primers used for cloning and mutagenesis of EPAC1 are detailed in Supplementary Table 1.

### Cytosol-nuclei cell fractionation
Cells plated into 10 cm petri dishes and confluent 80–90% were collected and incubated for 10 min on ice in hypotonic buffer (20 mM Tris-HCl (pH 7.4), 10 mM KCl, 2 mM MgCl$_2$, 1 mM EGTA, 0.5 mM DTT, 0.3% NP-040), followed by centrifugation at $1000 \times g$ at 4 °C for 5 min to separate the nuclei (pellet) and cytoplasm (supernatant). The pellet was washed twice with isotonic buffer (20 mM Tris-HCl (pH 7.4), 150 mM KCl, 2 mM MgCl$_2$, 1 mM EGTA, 0.5 mM DTT) resuspended and incubated in cold RIPA buffer (5 mM Tris-HCl (pH 7.4), 150 mM NaCl, 0.1% SDS, 0.5% sodium deoxycholate, 1% NP-40) for 10 min on ice. Samples were then centrifuged at $2000 \times g$ (4 °C) for 3 min and supernatant was collected as nucleic lysate. All buffers were supplemented with cOmplete™ Protease Inhibitor Cocktail (Roche Diagnostics) and PhosSTOP™ phosphatase inhibitor cocktail (Roche Diagnostics).

### Western blotting
Cells were lysed in cold RIPA buffer supplemented with protease and phosphatase inhibitors. Lysates (20–30 μg of the single fractions) were loaded onto 4–12% precast polyacrylamide gel (Bolt 4–12%, Bis-Tris plus gels; Thermo Fisher Scientific) for electrophoresis and run at 100 V. Proteins were then transferred onto polyvinylidene fluoride (PVDF) membranes (Thermo Fisher Scientific), and blocked for 1 h at room temperature in 10% (w/v) non-fat-dry milk-Tris-buffered saline/Tween 20 (0,1%) (TBST). The membranes were then incubated overnight at 4 °C with continuous rotation with 1:1000 primary antibody in 5% normal milk-TBST. Membranes were washed three times with TBST at room temperature and incubated for 1 h at room temperature with 1:3000 peroxidase-conjugated secondary antibodies. Membranes were developed with enhanced chemiluminescence (Luminata Crescendo Western HRP, Merck Millipore) and imaged using an ImageQuant LAS 4000 mini system equipped with a CCD camera (LasAF 2.6.0 software). Antibodies used: anti-GAPDH (H12) (Santa Cruz Biotechnology, sc-166574); anti-Lamin A/C (4C11) (Sigma, SAB4200236); anti-EPAC1 (5D3) (Cell Signalling Technologies, 4155); anti-EPAC2 (D3P3J) (Cell Signalling Technologies, 43239); anti-Sumo2/3 polyclonal (Enzo Life Sciences, BML-PW9465); anti-GFP polyclonal (Thermo Fisher Scientific, A11122); anti-βActin (AC15) (Abcam, 49900). Secondary antibodies: Goat anti-Rabbit IgG (H + L) Secondary Antibody, Goat anti-Rabbit IgG (H + L) Cross-Adsorbed Secondary Antibody, Goat anti-Mouse IgG (H + L) Highly Cross-Adsorbed Secondary Antibody. For subsequent detections, membranes were stripped using

Restore Western Blot Stripping Buffer (Thermo Fisher Scientific, 46430) for 15 min at room temperature and then washed with TBST.

## Confocal live-cell imaging

Cells were plated on 15 mm glass coverslips coated with poly-L-Lysine (Sigma-Aldrich, P4707). Before imaging experiments, cells were rinsed twice with Ringer's modified buffer (NaCl 125 mM; KCl 5 mM; $Na_3PO_4$ 1 mM; $MgSO_4$ 1 mM; Hepes 20 mM; glucose 5.5 mM; $CaCl_2$ 1 mM; pH adjusted to 7.4 using 1 M NaOH) and mounted onto an open perfusion chamber RC-25F (Warner Instruments, Hamden, CT, USA). Indicated treatments were performed both acutely on stage or cells were pre-treated. Images were collected on a Leica SP5 Confocal scanning microscope using oil immersion 40x (HCX PL Apo lambda blue 40x/1.25 Oil UV, Leica, Wetzlar, Germany) or 60x (HCX PL Apo lambda blue 63x/1.40 Oil UV, Leica, Wetzlar, Germany) objectives. Data were collected using the LasAF 2.6.0 software and post-processed using FIJI (version 1.53c).

## Fluorescence resonance energy transfer (FRET)

HEK293 cells were plated onto glass coverslips and transfected with the FRET-based cAMP sensor EPACH187. Twenty-four hours after transfection cells were mounted onto an open perfusion chamber RC-25F (Warner Instruments, Hamden, CT, USA) and perfused using a homemade gravity-fed perfusion system. The cells were bathed in Ringer's modified buffer (NaCl 125 mM; KCl 5 mM; $Na_3PO_4$ 1 mM; $MgSO_4$ 1 mM; Hepes 20 mM; glucose 5.5 mM; $CaCl_2$ 1 mM; pH adjusted to 7.4 using 1 M NaOH). The experiments were performed on an Olympus IX81 inverted microscope (Olympus, Tokyo, Japan) equipped with a beam-splitter (Dual-ViewTM, Optical Insights, Santa Fe, New Mexico, NM, USA) and a CCD camera (F-ViewII, Soft Imaging System, Münster, Germany). The cyan fluorescent protein (mTurquoise) was excited for 200 milliseconds at 430 nm, while the emission fluorescence was collected every 10–15 s for both donor (mTurquoise) and acceptor (Venus) fluorophores at 480 and 545 nm, respectively. Automatic image collection and preliminary analysis were performed using the Cell R software version Cell R 2.2 (Olympus, Tokyo, Japan) and then analyzed with ImageJ plugin (version 1.53c). Raw data were transferred to Excel (Microsoft, Redmond, WA, USA) for background subtraction and generation of the ratios.

## Fluorescence recovery after photobleaching (FRAP)

HEK293 cells were plated onto glass coverslips and transfected with EPAC1-YFP. Twenty-four hours after transfection cells were mounted onto an open perfusion chamber RC-25F (Warner Instruments, Hamden, CT, USA) and bathed in Ringer's modified buffer (NaCl 125 mM; KCl 5 mM; $Na_3PO_4$ 1 mM; $MgSO_4$ 1 mM; Hepes 20 mM; glucose 5.5 mM; $CaCl_2$ 1 mM; pH adjusted to 7.4 using 1 M NaOH). FRAP experiments were performed on a Leica SP5 Confocal microscope using a 488-nm laser. Bleaching was performed using 100% laser power for 4 cycles, and images were collected every 200–300 ms. Images were collected using the LasAF 2.6.0 software, fluorescence intensity at the bleached spot and of the whole cell was measured using the FIJI (version 1.53c) plugin FRAP Profiler. Values are reported relative to the whole cell to control for photobleaching during acquisition.

## Protein expression and purification

Epac1-WT recombinant protein was expressed as MBP-fusion protein (pET24b) in the BL21 Rosetta E. coli strain. The cells were grown in LB media (ThermoFisher) containing 50 mg/ml kanamycin and 34 mg/ml chloramphenicol to an OD600 of 0.6, induced with 0.4 mM IPTG and incubated overnight at 25 °C. Cells were resuspended in Lysis buffer (500 mM NaCl, 25 mM Tris-HCl (pH 8.5), 15 mM Imidazole, 1 mM PMSF, 1 mg/ml Leupeptin, 1 mg/ml Aprotin and 1 mg/ml Pepstatin) and lysed in a High-Pressure Homogenizer (Homogenising Systems LTD). After centrifugation (126,000 × g, rotor Beckman Ti70, 40 min, 4 °C) the

supernatant was loaded onto an HisTrap FF Crude column (Cytiva). The protein was washed three times with equilibration buffer (500 NaCl, 25 mM Tris-HCl (pH 8.5)) containing 16 mM, 32 mM, and 48 mM Imidazole, respectively, and eluted with equilibration buffer containing 400 mM Imidazole. Subsequently, HRV-3C Protease cleavage (in presence of 1 mM 2-Mercaptoethanol and 1 mM EDTA (pH 8)) was performed overnight at 4 °C at the ratio 1:50 protease:protein, followed by batch affinity purification using an Amylose Resin to remove the MBP and any uncleaved fusion protein. The Flow-through was concentrated by Centrifugal concentrator (Amicon® Ultra-15 Centrifugal Filter Unit, 50,000 NMWL, Merck Millipore) and applied to a Superdex 200 increase 100/30 column (Cytiva). Pure protein is eluted with "High Salt buffer" (500 mM NaCl, 25 mM Tris-HCl (pH 8.8) and 5 mM $MgCl_2$).

## Phase separation measurements

Purified recombinant Epac1-WT (50 µM) in a High Salt buffer (500 mM NaCl, 25 mM Tris-HCl (pH 8.8), and 5 mM $MgCl_2$) was diluted using a dilution buffer (25 mM Tris-HCl (pH 8.8) and 5 mM $MgCl_2$) to 5, 10, 15, 20, and 25 µM concentration corresponding to final NaCl concentrations of 50, 100, 150, 200, and 250 mM, respectively, with and without 30 µM cAMP. Then, the solutions were incubated at 20 °C for 15 min and cleared by centrifugation (16,900 × g, 30 min, 20 °C). Subsequently, Epac1-WT concentrations in the cleared supernatant were measured through absorbance at 280 nm using a Nanodrop Spectrophotometer (DeNovix DS-11). Each Epac1-WT residual concentration was, then, plotted in function of the salt concentration to determine the threshold of phase separation.

## DNA fluorescence in situ hybridization (FISH)

Cells were plated onto glass coverslips and transfected with EPAC1-YFP. After 24 hours cells were fixed with 4% PFA in PBS for 10 min and washed 3 times with PBS. Cells were dehydrated by serial incubations in ethanol 70%, 85%, and 100% for 1 min at room temperature. Probe hybridization mixture was prepared by mixing 7 µl of FISH Hybridization Buffer (Agilent G9400A), 1 µl of FISH probe (see below) and 2 µl of water. Five microliters of the mixture were added on a slide and the coverslip was placed on top. Coverslips were sealed with rubber cement. Denaturation was performed at 78 °C for 15 min and slides were incubated at 37 °C in the dark overnight. The coverslip was then incubated in pre-warmed wash buffer 1 (0,4X SSC; 0.3% NP40 pH 7.5) at 73 °C for 2 min, and in wash buffer 2 (2X SSC; 0,1% NP40 pH7) for 1 min at room temperature. Coverslips were air-dried, mounted on slides using Vectashield (Vector Laboratories), and sealed with nail polish. Images were acquired on a Zeiss LSM900 with Airyscan confocal microscope in modality super-resolution with a 63× objective using the ZEN Blue 3.0 software and processed using FIJI (version 1.53c). A specific DNA FISH probe for chromosome 6p22.2 was custom-designed and generated by Agilent to target Histone locus 1. The design input region was chr6:26019490-26201715 (182.226 kb) and the design region was chr6:26019341-26201862 (182.522 kb). While for the control experiments, we used 3 different commercial DNA FISH probe: chromosome 21 (Vysis LSI21 08L54-020) approximately 220 kb 21q22.13-q22.2 (chr21: 39439949-39659711), centromeric region of chromosome 15 (15p11.1q11.1) (Vysis D15Z4; 06j36-015) and centromeric region of chromosome 12 (12p11.1-q11.1) (Vysis 30-160012).

## Immunofluorescence

Cells were plated on coverslips coated with poly-L-Lysine (Sigma-Aldrich, P4707) and transfected with EPAC1-YFP or its mutant versions. Twenty-four hours after transfection cells were fixed using 4% paraformaldehyde (PFA) (Santa Cruz Biotechnology, sc-281692) in PBS for 10 min. After washing cells 3 times in PBS, the coverslips were put into

a humidifying chamber for subsequent steps. Cell permeabilization was performed using 0.2% Triton X100 (Serva, 39795.02) in PBS for 10 min, followed by 3 PBS washes. After blocking with 2% BSA for 1 h cells were incubated with the indicated primary antibodies at a concentration of 1:300 in PBS except for endogenous EPAC1 that was used 1:50. The antibodies used for immunofluorescence were SMN (F-5) (Santa Cruz Biotechnology, sc-365909); Nucleolin polyclonal (Sigma-Aldrich, N2662); NPAT (27) (Santa Cruz Biotechnology, sc-136007); PML (Sigma-Aldrich, PLA-0172); EPAC1-488 (EPR1672) (Abcam, ab201506); Sumo 2/3 polyclonal (Abcam, ab3742); RanBP2 (D-4) (Santa Cruz Biotechnology, sc-74518). Secondary antibodies: Alexa Fluor 568 (Invitrogen, A-11011); Alexa Fluor 647 (Thermo Scientific, A-21244); Alexa Fluor Plus 647 (Thermo Scientific, A32728).

## RNA sequencing

HEK293 cells were seeded on 10 cm petri dishes and transfected with either EPAC1-YFP or its mutant version EPAC1Δ2-148-YFP. After 24 h cells were treated with DMSO (vehicle control) or 8-pCPT-2′-O-Me-cAMP-AM (5 μM) for 40 min. Cells were then FACS-sorted (FACS facility Veneto Institute of Molecular Medicine, Padua, Italy) using the YFP fluorescence. Total RNA was extracted using RNeasy Mini Kit (Qiagen ID: 74104) complemented with on-column DNAse digestion with the RNAse-free DNAse set (Qiagen ID: 79254) according to the manufacturer's instructions. RNA was quantitatively and qualitatively evaluated using NanoDrop 2000c (Thermo Fisher Scientific) and Agilent Bioanalyzer 2100 (Agilent Technologies), respectively.

RNA-seq libraries were prepared from 1 μg of total RNA, using the Illumina's TruSeq Stranded Total RNA Sample Preparation Kit (Illumina, San Diego, CA, USA), according to the manufacturer's protocol. cDNA libraries were qualitatively checked on the Bioanalyzer 2100 and quantified by fluorimetry using the Quant-iTTM PicoGreen® dsDNA Assay Kit (Thermo Fisher Scientific) on NanoDrop™ 3300 Fluorospectrometer (Thermo Fisher Scientific). Sequencing was performed on NextSeq500 platform, generating for each sample almost 100 M of 100 bp paired-end reads.

Raw RNAseq reads were initially inspected by FASTQC (https://www.bioinformatics.babraham.ac.uk/projects/fastqc/) and, then, low-quality regions and adapters were removed using fastp[69]. Cleaned reads were aligned onto the human genome (version hg19) by the ultrafast STAR program[70] providing a list of splice junctions from Gencode[71]. Reads mapping on known human genes were counted using FeatureCounts[72] and differential gene expression was calculated using CuffDiff[56]. Only genes showing $p$-values < 0.01 were selected for downstream analyses.

## Real-time qPCR

Expression levels of HIST1H2AI gene was analyzed using real-time, quantitative PCR as a validation of RNAseq results. RNA was extracted from HEK cells expressing EPAC1-YFP using the RNeasy® mini kit from Qiagen and quantitatively evaluated using NanoDrop 2000c (Thermo Fisher Scientific). Reverse transcription was performed using the SuperScript™ VILO™ cDNA Synthesis Kit (Invitrogen, 11754050). The cDNA samples were diluted 1 to 10 and used in triplicates for multiplex qPCR with TaqMan™ Fast Advanced Master mix (Applied Biosystems 4444557) and fluorescent probes 18 s (VIC primer-limited; Assay ID: Hs99999901_s1) and HIST1H2AI (FAM; Assay ID: Hs00361878_s1). The thermal cycling conditions were composed of 50 °C for 2 min followed by an initial denaturation step at 95 °C for 20 s then 45 cycles at 95 °C for 1 s, 60 °C for 20 s. The relative quantification in gene expression was determined using the 2-ΔΔCt method. Using this method, we obtained the fold changes in gene expression normalized to an internal control gene, and relative to one sample (calibrator). The ribosomal RNA 18 s gene was used as an internal control to normalize all data and the untreated DMSO sample was chosen as the calibrator in each set of experiments.

## Cell cycle analysis

HEK293 cells were cultured at 37 °C under a 5% $CO_2$ atmosphere. For synchronization at the G1/S border a double thymidine-block (DTB) protocol was used. Briefly, cells were treated with 2 mM thymidine for 16 h and released for 8 h in fresh medium before the second thymidine block was performed for another 16 h. Still in thymidine block, cells were treated for 1 h with DMSO or 8CPT and then were either collected directly or released in fresh medium and harvested after 1 h or 8 h. For the flow cytometric analysis, cells were fixed in cold 70% ethanol for 45 min and then spun down, washed in PBS 1X, and incubated in the propidium iodide (PI) solution (50 μg/ml PI, 0.2 mg/ml RNase A and 1× PBS) for 1 h at RT in the dark. Incorporated PI and associated fluorescence are proportional to the amount of cellular DNA. Cell cycle progression was analyzed using BD FACSCanto II system and data were analyzed using BD FacsDiva software 8.0.

## Analysis of the descriptor circularity index of single nuclear condensates

Images were acquired with the confocal microscope Zeiss LSM900 equipped with Airyscan 2 – Super Resolution System and an oil immersion 63X objective (Zeiss, Zeiss Plan-Apochromat 63X/1.4 Oil) using the software Zeiss Efficient Navigation Blue (ZEN Blue 3.0, Zeiss). Once processed, images were analyzed with Fiji (ImageJ2 version 1.53c, National Institute of Health, USA). In the case of z-stacks, a z-projection was obtained averaging the fluorescence intensity. To select only nuclear condensates, measurements were performed inside nuclear ROIs based on DAPI staining of DNA. To remove the background and to ensure an analysis of true condensates, a threshold was automatically applied and particles with a surface lower than 20 px² were excluded. Resulting images were converted to masks and the Watershed Separation method was used to segment particles artificially fused during the z-projection. The circularity index of 1595 particles was analyzed.

## Colocalization analysis, 3D volume rendering

Images were acquired with the confocal microscope Zeiss LSM900 equipped with Airyscan 2 – Super Resolution System and an oil immersion 63X objective (Zeiss, Zeiss Plan-Apochromat 63X/1.4 Oil) using the software Zeiss Efficient Navigation Blue (ZEN Blue 3.0, Zeiss). A maximum z-projection of z-stacks images was obtained averaging the fluorescence intensity and to select only nuclear condensates, measurements were performed inside nuclear ROIs based on DAPI staining of DNA. To consider spatial exploration of the colocalization signals in immunofluorescence experiments object-based analysis was performed using FIJI (version 1.53c) JaCoP plugin. To avoid under- or over-estimation of colocalization events, both distance between geometrical centers of objects (including the total shape of the structures) and center of mass-particles coincidence were considered for green versus red channels. This approach considers positive colocalization if the distance between the objects is below the optical resolution. The average of positive colocalization events was then normalized against the average number of each type of condensate per nucleus (PML = 16, NPAT = 6, SMN = 4) and represented with the respective error bars (SD).

3D volume rendering for each type of colocalizing condensates (PML, SMN and NPAT) was performed using FIJI (version 1.53c) VolumeJ plugin. Z-stacks for each channel with minimal manual intensity correction was used to generate a 3D reconstruction. Images were then cropped around the respective colocalizing events.

As for FISH experiment datasets, images were analyzed with FIJI (version 1.53c). FISH foci were identified in maximum z-projections and

the x and y coordinates were used as reference points to guide the automatic detection of either overlapping or contiguous foci. Manual minimal thresholds were chosen for the immunofluorescence channels. For each FISH experiment a plug-in of the colocalization heatmap was used on the obtained images to calculate the normal mean derivation product (nMDP) for each pixel. The generated heatmap shows the positive (shown as warm colors (yellow-red)) and negative (shown as a cold color (greenish-blue, blue)) correlations in pixels to visualize the colocalization of two objects tagged with different probes. Plugin also allows quantitative analysis by calculating the index of correlation (Icorr). The index represents fraction of positively correlated (colocalized) pixels in the analyzed images. Icorr values for each probe were then averaged and represented with the respective error bars (SD). The experiments were repeated at least three times and the number of the FISH foci considered for each probe are the following: Chr6p22.2 = 110; CEP15 = 37; Chr21q = 34 CEP12 = 10.

## Statistics
Statistical analysis was performed using Graphpdad Prism (Version 8.0.2). All experiments were repeated at least three times (unless stated otherwise) and data were presented as mean ± SD (unless stated otherwise). No data were excluded from the analyses (unless stated otherwise). The statistical significance between two groups was calculated by unpaired two-tailed Student's t-test. The statistical significance between multiple groups was calculated by one-way ANOVA.

## Reporting summary
Further information on research design is available in the Nature Portfolio Reporting Summary linked to this article.

## Data availability
The RNAseq data generated in this study have been deposited in the public archive of high throughput sequencing data SRA (Sequence Read Archive) under accession number PRJNA1001149. All uncropped gels and numerical values are provided in the Source data file. The accession number of the Human reference genome used in this study is GRCh38.p13. Source data are provided as a Source data file. Source data are provided with this paper.

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

## Acknowledgements

We thank Prof. Xiaodong Cheng (Department of Integrative Biology & Pharmacology Texas Therapeutics Institute University of Texas Health Science Center) for EPAC1-YFP and EPAC1$^{\Delta2-148}$-YFP constructs. We thank Prof. Graziano Martello and Dr. Alessandro Carrer for reading the manuscript. We thank Prof. Luca Scorrano (University of Padua), Dr. Alessandra Dall'Agnese (Whitehead Institute), Prof. Richard A. Young (Whitehead Institute) and Prof. Jin Zhang (UCSD) for helpful discussions and suggestions and the members of the Lefkimmiatis laboratory for

helpful discussions. We thank Dr. Adina Cordella for help and support with FISH experiments. Funding for the RNAseq experiments was obtained from the MIUR - National Research Program (PNR) Flagship InterOmics Project (cod. PB05) to L.M. The in vitro biochemical experiments were supported by 'Programma per Giovani Ricercatori - Rita Levi Montalcini 2016' granted by the Italian Ministry of Education (Project number: PGR16HTPSF) to M.L. This work was supported by grants from the Foundation "Cassa di Risparmio di Padova e Rovigo" (CA.RI.PA.RO) (SIGMI), the Italian Ministry of University and Scientific Research (PRIN 2017), Human Frontier Science Program Research Grant (HSFP ref: RG0024/2022) and the AIRC foundation for cancer research (grant IG2021 ID 26140) to K.L.

## Author contributions

LF.I., D.B., F.C., NC.S., G.DB., D.M., C.A., M.L., M.V., and F.G. performed the experiments. E.P. and AM.D. performed and analyzed the RNAseq experiments, L.M., G.P., and K.L. funding acquisition for the RNAseq experiments. LF.I., N.C.S., M.L., F.C., G.D.B., and M.V. contributed to the scheme and figures. D.B. and L.S. performed FISH experiments. K.L. prepared the original draft. L.F.I. and G.D.B. contributed to the revision of original draft. K.L. and L.F.I. conceived the idea and designed the experimental plan. All the authors discussed the results and commented on the manuscript.

## Competing interests

The authors declare no competing interests.
