## [Peer Review File · Nature Communications]

Cyclic AMP induces reversible EPAC1 condensates that regulate histone transcriptionREVIEWER COMMENTS

Reviewer #1 (Remarks to the Author):

The manuscript by Lannucci et al investigated and reports an unexpected mechanism underlying cAMP-induced nuclear condensation of EPAC1 and identification of the primary functional target regulated by nuclear EPAC1 condensates. Using various in vitro approaches, the authors found specific mechanism regulating the entry of EPAC1 into the nucleus, dependent of intrinsically disordered regions present at its amino-terminus. They demonstrate that nuclear EPAC1 condensates assemble at genomic loci on chromosome 6 and promote the transcription of a histone gene cluster. Overall, the experiments were well designed and executed. The results are important given the phenomenon of cAMP-induced nuclear condensation of EPAC was reported. Especially the results about the mechanism underlying EPAC1 entering into nucleus are convincing and support two distinct regions for engaging the nuclear pore and entering. However, I have three major concerns raised by the information reported in a new publication (Yang et al, Science Advances April 20, 2022. DOI: 10.1126/sciadv.abm2960) about the functional roles of cAMP-induced EPAC1 nuclear condensates in cell models of HUVEC and HEK.

1. Using high throughput whole transcriptome RNA-sequencing, the authors were focusing on identification and characterization the functional role of nuclear EPAC1 condensates on histone gene cluster. Yang et al reported Activation of Epac1 by intracellular cAMP triggers phase separation and the formation of nuclear condensates containing Epac1 and general components of the SUMOylation machinery to promote cellular sumoylation (DOI: 10.1126/sciadv.abm2960). Given that histone sumoylation is associated with regulation on transcriptions (Shiio Y, et al, PNAS 2003;100:13225<https://doi.org/10.1073/pnas.1735528100>), the authors need to address and discuss the potential role of histone sumoylation in EPAC1 nuclear condensate-regulated transcription.

2. Now, the authors should address or discuss the differences in all presented scientific data between this manuscript and the Yang et al reported one, demonstrating the novelty and deeper insight into similar topic.

3. In mammals, the EPAC protein family contains two members: EPAC1 and EPAC2. Both EPAC isoforms function by responding to increased intracellular cAMP levels in a PKA-independent manner and act on the same immediate downstream effectors, the small G proteins Rap1 and Rap2. EPAC1 is the major isoform in HUVECs, while both are detected in HEK293 cells. The author should discuss their claimed unexpected information in the context of EPAC2, in particular within their cell model using HEK293.

Minor comment:

1. Line 21, after stating “accumulating evidence”, there is only one citation of reference 6.
2. Line 34, please extend description by more information about “the classic model”.
3. Line 43, correct the typo “that PKA it not...”
4. Line 51, please correct grammar error “ which effectively enters...”

5. Line 70 “we overexpressed EPAC1-YFP in EPAC1-deficient HEK cells”. There is critical technical information missing about “EPAC1-deficient HEK cells”. Do the authors use knock-out or knock-down technology?

6. Line 398, why was there such big range 4 to 10 to select image fields between different experiments.

7. Line 400, please describe in detail the standard to define as “selected nuclear ROIs”.

Reviewer #2 (Remarks to the Author):

The manuscript by Iannucci et al addresses the role of EPAC1 in response to cyclic AMP and describe a previously uncharacterized function in the regulation of histone transcription. This regulation involves the ability of EPAC1 to undergo liquid-liquid phase separation. These experiments have the potential to provide new insights on how cAMP might regulate transcription. However, despite this potential impact, this manuscript suffers of several shortcomings.

1. The claim that EPAC can phase separate remains preliminary. The experiments performed by the authors have often been used in the literature as evidence of phase separation but they are very limited and only show consistency. Several claims that the authors made should be quantified. For example, it is not immediately clear from the images that the condensates are indeed spherical. Similarly, the paper could benefit from experiments showing that fusing condensate regain sphericity quickly. While a definite demonstration that condensate indeed form by LLPS is a difficult task, this paper could be strengthened by at the very least some quantifications. At this point, the claim is not strongly supported.

2. The overlap between the EPAC and histone locus and the histone locus body is not easy to interpret mechanistically. The EPAC condensate appears to colocalize but not perfectly with the histone locus on chromosome 6. The colocalization in Figure 4 with NPAT appears stronger. However, there is only one example shown and not quantification. EPAC condensate are not passive followers of HLBs, as in the absence of cAMP stimulation there are no condensate but HLBs are still present. Moreover, the EPAC condensates do not always overlap with the HLBs. It would be important to understand these relationship in quantitative term and their possible hierarchy. For example, it is unclear if HLBs are required for EPAC condensate to overlap with the histone locus. Experiments to understand the relationship between the two biocondensates would be crucial.

Minor point:

Recent work has shown that the HLB forms by phase separation (Hur et al Dev Cell 2020) and has elucidated ways to perturb this process. Addressing the effects of some of these perturbations on EPAC biocondense could be useful.

Reviewer #3 (Remarks to the Author):

****Summary****

The overarching goal of this work by Iannucci et al. is to uncover the mechanism of PKA-independent nuclear cAMP signaling. To this end, the authors focus on the cAMP-activated GEF EPAC1 that is known to shuttle from the cytoplasm into the nucleus upon cAMP production. This work nicely combines mutagenesis and chemical biology to show that nuclear EPAC1 localizes to cAMP-induced and -dependent foci. The authors furthermore intriguingly link EPAC1 foci localization to differences in gene expression, most notably of histone gene clusters on chromosome 6. The authors conclude that EPAC1 forms nuclear condensates in response to PKA-independent cAMP signaling and that this cAMP/EPAC1 condensate axis represents a novel signaling paradigm. Overall, this work is interesting and thought-provoking, and in principle a nice fit for Nature Communications. However, the authors over-interpret their results throughout the manuscript, raising doubts that EPAC1 is indeed a central hub in PKA-independent cAMP signaling. Thus, this work requires major revisions.

****Major comments****

Specifically, the burden of proof has not been met for three key statements:

- 1.) The observed co-localization of EPAC1 Δ 179-208-YFP and RANBP2 in Extended Data Figure 1D is not sufficient to draw the conclusion that they directly bind each other, especially given that: (a) this is a novel EPAC1 mutation, (b) nuclear localization signals (NLSs) typically bind to nuclear transport receptors (NTRs) and not nucleoporins like RANBP2, and (c) there is prior biochemical evidence for a direct, NLS/NTR-independent tethering of EPAC1 to RANBP2 (Gloerich et al. 2011). Thus, to truly identify the specific mechanism of EPAC1 nuclear import (as claimed in lines 68-69), the authors must directly test binding of wild-type and mutant EPAC1 to NTRs and nucleoporins, for example by immunoprecipitation assays (IPs) or binding assays with purified components.
- 2.) The data presented is not sufficient to definitively conclude that EPAC1 forms condensates, as opposed to being merely recruited to condensates. In fact, the authors show evidence for the latter in the form of (a) the (incomplete) co-localization of EPAC1 foci, PML bodies and HLBs, and (b) changes in histone gene cluster transcription between wild-type and mutant EPAC1. To show that EPAC1 is indeed the core scaffold component of a novel type of condensates (that unequivocally can be called EPAC1 condensates), the authors must reconstitute the phase separation of EPAC1 in vitro.

3.) The author's efforts to map the domains driving the potential phase separation of EPAC1 are cursory, leaving more doubts than providing understanding for three main reasons. First, IDRs are by no means cardinal to phase separation as claimed in line 126. What is cardinal to phase separation is multivalency, which can be achieved through both folded and disordered domains. Indeed, the prevalent view in the field is that IDR typically work in concert with oligomerization domains, and that IDRs often merely tune phase separation behavior (see e.g. Martin and Holehouse, 2020). Second, EPAC1 appears to only have one very short IDR (approx. residues 1-66) based on DisoPred3, NetSurfP2 and AlphaFold2 predictions. The other IDRs that Iannucci et al. highlighted merely seem to be classic disordered linkers connecting folded domains. Third, it appears that EPAC1 contains folded domains like the DEP domain that are known to mediate protein-protein interactions in signaling pathways. In this regard, the author note that truncating the DEP domain (by virtue of the EPAC1 Δ 48-148 mutant) interferes with condensate localization. Together, the domain mapping appears biased towards pushing the preconceived model that EPAC1 undergoes IDR-dependent liquid-liquid phase separation, whereas both the data presented in this manuscript as well as the literature cast doubts on this model. Thus, the authors must devise more comprehensive and unbiased experiments to distinguish between the EPAC1 phase separation and condensate recruitment models (and identify the domains that drive this), or significantly revise and tone down their conclusions.

****Minor comments****

- How were the EPAC1-deficient HEK cells generated? This information is missing in the manuscript, despite the importance of this resource for this study.

- Lines 115-118: Imprecise argument. Hexanediol sensitivity is not diagnostic of LLPS, but merely suggests that hydrophobic interactions may be relevant. Much stronger evidence for the model that EPAC1 behaves as if in a viscous liquid phase comes from the fusion and FRAP experiments.

- It would greatly help if the authors would consolidate the EPAC1 domain architecture cartoons in Figure 3A and Extended Data Figure 1A into one scheme.

- Structuring the manuscript into sections with subheadings would greatly help the flow of the paper.

Reviewer #4 (Remarks to the Author):

In this manuscript, Iannucci et al. explored the nuclear functions of cAMP effector, EPAC1. They found that upon cAMP elevation, EPAC1 enters into the nucleus where it forms reversible biomolecular condensates through liquid-liquid phase separation. This phenomenon is independent on the canonical effector of cAMP, PKA. Moreover, they found that EPAC1 condensates assemble on chromosome 6 with colocalization with histone gene regulator NPAT to promote histone gene transcription. Overall, this is an interesting story as it uncovered a mechanism through which cAMP contributes to nuclear spatial compartmentalization and promotes the transcription of specific genes. However, lacking of proper controls and highly rely on cell imaging make some conclusions not convincing. The following are my concerns that need addressed.

1. My big concern is whether and how the EPAC1 condensate regulates histone gene expression. It is well-known that histone gene expression is tightly controlled by cell cycle. It is unknown whether treatment with cAMP agonist will affect cell cycle. The authors should try to explain how EPAC1 condensates regulate histone gene expression. What about the EPAC1 mutants? Does the EPAC1 732-764 mutant and EPAC1179-208 mutant that cannot enter into the nucleus regulate histone gene expression independent on cell cycle?

2. Fig. 4: The co-localization of NPAT with EPAC1 condensates is not that clear. Moreover, it is recommended to use other techniques, such as Co-IP to confirm the interaction between NPAT and EPAC1. ChIP EPAC1 and NPAT at histone locus is also a better way to demonstrate that point.

3. To make the story more convincing, the authors are recommended to express and purify the recombinant EPAC1 to demonstrate it can form liquid-liquid phase separation in vitro.

4. The probe signal for chromosome 21 is very weak. It is too preliminary to conclude that there is no co-localization between chromosome 21 and EPAC1 condensates given the so weak signals.

5. Extended data Fig. 1: The authors should examine whether these two domains affect the protein expression of EPAC1. For extended data 1c and 1d, WT EPAC1 should be added as a positive control.

Other comments:

6. Fig. 1c: what is time points used for calculation?

7. Fig. 1d and 1g: The DAPI should be added to indicate the nucleus.

8. Fig. 5b: the EPAC1 1-148 mutant should be included to show its effect on histone gene expression.

9. The language of this manuscript needs to be polished. There are a lot of grammar errors and mistakes.

Point-by-point response to the reviewers' comments

Firstly, we would like to thank the reviewers for taking the time to review our manuscript and their insightful comments. In the present version we added more experiments solving the concerns expressed. Hereafter a point-by-point response is presented.

Reviewer #1 (Remarks to the Author):

The manuscript by Iannucci et al investigated and reports an unexpected mechanism underlying cAMP-induced nuclear condensation of EPAC1 and identification of the primary functional target regulated by nuclear EPAC1 condensates. Using various in vitro approaches, the authors found specific mechanism regulating the entry of EPAC1 into the nucleus, dependent of intrinsically disordered regions present at its amino-terminus. They demonstrate that nuclear EPAC1 condensates assemble at genomic loci on chromosome 6 and promote the transcription of a histone gene cluster. Overall, the experiments were well designed and executed. The results are important given the phenomenon of cAMP-induced nuclear condensation of EPAC was reported. Especially the results about the mechanism underlying EPAC1 entering into nucleus are convincing and support two distinct regions for engaging the nuclear pore and entering. However, I have three major concerns raised by the information reported in a new publication (Yang et al, Science Advances April 20, 2022. DOI: 10.1126/sciadv.abm2960) about the functional roles of cAMP-induced EPAC1 nuclear condensates in cell models of HUVEC and HEK.

Major points

1. Using high throughput whole transcriptome RNA-sequencing, the authors were focusing on identification and characterization the functional role of nuclear EPAC1 condensates on histone gene cluster. Yang et al reported Activation of Epac1 by intracellular cAMP triggers phase separation and the formation of nuclear condensates containing Epac1 and general components of the SUMOylation machinery to promote cellular sumoylation (DOI: 10.1126/sciadv.abm2960). Given that histone sumoylation is associated with regulation on transcriptions (Shiio Y, et al, PNAS 2003; 100:13225 <https://doi.org/10.1073/pnas.1735528100>), the authors need to address and discuss the potential role of histone sumoylation in EPAC1 nuclear condensate-regulated transcription.

We agree with the reviewer that the involvement of EPAC1 condensates in SUMOylation raises the question of whether this process is at the basis of the transcriptional effects we observe on the histone locus 1. As demonstrated by the manuscript suggested by the reviewer¹ and other literature², the process of SUMOylation is mostly connected to transcriptional repression, while on the contrary, we observe increased transcription of histone genes in response to nEPAC1 condensate formation. Nevertheless, to address the reviewer's concern, we performed ad hoc experiments.

- We tested the ability of nEPAC1 condensates to activate transcription of histones in the presence or absence of a SUMOylation inhibitor. As shown in the new **Extended Data Figure 7**, a SUMOylation inhibitor that was fully able to completely reverse the effect of EPAC1 activation on SUMOylation, had no effect on histone transcription assessed by real time experiments on a representative histone gene.

- In addition, we performed immunofluorescence experiments simultaneously labelling histone locus bodies (NPAT) and SUMO2/3 in cells expressing EPAC1-YFP. Interestingly, we observed colocalization of EPAC1 with NPAT and SUMO2/3 but we never found a complex containing EPAC1 NPAT and SUMO2/3 **Extended Data Figure 7a,b**. In the new version of the manuscript, we present the new data and discuss the implications of EPAC1 condensates on SUMOylation and transcription.

2. Now, the authors should address or discuss the differences in all presented scientific data between this manuscript and the Yang et al reported one, demonstrating the novelty and deeper insight into similar topic.

The manuscript from Yang et al. was published while our manuscript was in revision, and presents some similarities with our work, albeit not extensive. As requested by the reviewer we performed experiments that excluded the possible mechanistic overlap between our results and those of Yang et al. (please see answer to previous point), these data were also further elaborated in the discussion. Other than experimentally demonstrating no mechanistic overlap, our manuscript offers a more detailed analysis of the mechanism through which EPAC1 enters the nucleus (**Extended Data Figure 1**) not at all addressed by Yang et al. Moreover, we present a more detailed map of the sequences necessary for the phase separation of EPAC1 and identify their importance in the recruitment of EPAC1 in condensates (**Figure 3 and Extended Data Figure 4 & 5**). Finally, we also offer a more detailed view of the modalities through which nEPAC1 condensates interact with different nuclear membraneless organelles, which revealed to be surprisingly organelle specific (**Figure 4f**). The more detailed analysis should be added to the completely unrelated and novel finding that EPAC1 condensates regulate histone transcription. Based on these considerations, in our opinion, Yang et al. does not affect the novelty of our findings. On the contrary, by reporting an unrelated and distinct function of EPAC1 condensates, Yang et al reinforces the need of more studies to better define how the cAMP/EPAC1 axis in the nucleus achieves its specificity of action and contributes to cell physiology.

3. In mammals, the EPAC protein family contains two members: EPAC1 and EPAC2. Both EPAC isoforms function by responding to increased intracellular cAMP levels in a PKA-independent manner and act on the same immediate downstream effectors, the small G proteins Rap1 and Rap2. EPAC1 is the major isoform in HUVECs, while both are detected in HEK293 cells. The author should discuss their claimed unexpected information in the context of EPAC2, in particular within their cell model using HEK293.

Contrary to EPAC1, that was shown to localize to the nuclear envelope, EPAC2 presents no nuclear or perinuclear localization and is mainly found (in a isoform-dependent manner) in the cytosol or plasma membrane³. In addition, the cell permeable cAMP agonist used for the

expression experiments was shown to be a better activator for EPAC1 as compared to EPAC2⁴ (suggesting that the transcriptional effects were most likely due to the activation of the former). Nevertheless, we agree with the reviewer that the structural similarity of the two proteins and the existence of common pathways between the two, dictate further investigation. For this reason, in the present version of the manuscript, we performed experiments addressing the subcellular localization of EPAC1 and its behavior in response to cAMP-dependent activation. In the new **Extended Data Figure 3a**, Western Blotting experiments demonstrate that EPAC2 was not found in the nuclear fractions of HEK cells. Moreover, in **Extended Data Figure 3b-c**, imaging experiments demonstrate that exogenously EPAC2 does not enter the nucleus and does not participate in the formation of any structures in response to the activation of the cAMP signalling cascade.

Minor points:

1. Line 21, after stating “accumulating evidence”, there is only one citation of reference 6.

In the *Nature Communications* format, it is not allowed to add references in the abstract therefore we had to take away reference 6. Nevertheless, the point on the evidence suggesting that PKA is not the only effector of cAMP in the nucleus is further discussed in the new introduction section and 3 references (12–15) have been added.

2. Line 34, please extend description by more information about “the classic model”.

In the new version we further elaborate on the classic model of the modalities through which PKA can be activated in the nucleus.

3. Line 43, correct the typo “that PKA it not...”

Done

4. Line 51, please correct grammar error “ which effectively enters...”

This phrase has been replaced in the new version of the manuscript.

5. Line 70 “we overexpressed EPAC1-YFP in EPAC1-deficient HEK cells”. There is critical technical information missing about “EPAC1-deficient HEK cells”. Do the authors use knock-out or knock-down technology?

The HEK cell line is well documented to be naturally EPAC1 deficient⁵, likely due to hypermethylation. The lack of EPAC is demonstrated in the manuscript in **figure 1a** by Western Blotting. In the new version of the manuscript, we make a better point on the deficiency of EPAC1 in HEK cells.

6. Line 398, why was there such big range 4 to 10 to select image fields between different experiments.

The lower number of cells depended on the low availability of the probe for the Chromosome 21. In the new version of the manuscript, we repeated these experiments and applied new probes as well (**Figure 7**).

7. Line 400, please describe in detail the standard to define as “selected nuclear ROIs”.

Done

Reviewer #2 (Remarks to the Author):

The manuscript by Iannucci et al addresses the role of EPAC1 in response to cyclic AMP and describe a previously uncharacterized function in the regulation of histone transcription. This regulation involves the ability of EPAC1 to undergo liquid-liquid phase separation. These experiments have the potential to provide new insights on how cAMP might regulate transcription. However, despite this potential impact, this manuscript suffers of several shortcomings.

Major points

1. The claim that EPAC can phase separate remains preliminary. The experiments performed by the authors have often been used in the literature as evidence of phase separation but they are very limited and only show consistency. Several claims that the authors made should be quantified. For example, it is not immediately clear from the images that the condensates are indeed spherical. Similarly, the paper could benefit from experiments showing that fusing condensate regain sphericity quickly. While a definite demonstration that condensate indeed form by LLPS is a difficult task, this paper could be strengthened by at the very least some quantifications. At this point, the claim is not strongly supported.

We thank the reviewer for the comments and suggestions. In the new version of the manuscript, we performed both the requested analysis and new experiments to further consolidate our claims on EPAC1 phase separation. In particular:

- As suggested by the reviewer we calculated the circularity index of EPAC1 condensates presented now in **Figure 2b**. In addition, using superresolution Airyscan microscopy we reconstructed condensates and both a rendering and a movie are presented (**Figure 2b** and **Extended Data Movie 3**).
- As suggested by the reviewer we calculated the circularity index of condensates before, during and after fusion and confirmed that after the expected momentaneous loss of circularity during fusion, the resulting condensate rapidly becomes spherical (**Figure 2d**).
- In the new version of the manuscript, we also tested the ability of purified EPAC1 to constitute condensates. We found that purified EPAC1 forms condensates in the presence of cAMP, while the mutant $\Delta 2-148$ was unable to form condensates independently of the cAMP levels. These data are now presented in **Figure 3c** and **Extended figure 4b-d**.

- In line with these data, we also tested the ability of the EPAC1 mutants $\Delta 2-148$, $\Delta 2-24$ and $\Delta 48-148$ to participate to condensates formed by EPAC1 wild type. We found that $\Delta 2-148$ was unable to participate, however both other mutants were able to participate in the growth of EPAC1 condensates albeit unable to trigger condensate formation by their own. These data are now presented in **Extended figure 5**.

2. The overlap between the EPAC and histone locus and the histone locus body is not easy to interpret mechanistically. The EPAC condensate appears to colocalize but not perfectly with the histone locus on chromosome 6. The colocalization in Figure 4 with NPAT appears stronger. However, there is only one example shown and not quantification. EPAC condensates are not passive followers of HLBs, as in the absence of cAMP stimulation there are no condensates but HLBs are still present. Moreover, the EPAC condensates do not always overlap with the HLBs. It would be important to understand these relationships in quantitative terms and their possible hierarchy. For example, it is unclear if HLBs are required for EPAC condensates to overlap with the histone locus. Experiments to understand the relationship between the two biocondensates would be crucial.

In the new version of the manuscript, we performed both, analysis and new experiments that better define the relation of EPAC1 condensates and HLBs.

- We performed quantification analysis of the overlap between EPAC1 condensates and other nuclear condensates (including HLBs). These data are now present in **Figure 4e** and demonstrate that EPAC1 and NPAT colocalize.
- In **figure 4f** we also present a 3D rendering of the interactions between nEPAC1 condensates and HLBs, PML and Cajal bodies which evidence clear differences on how the interaction between these organelles occurs.
- We further studied the relationship of EPAC1 condensates to NPAT and SUMO2/3. These experiments evidenced that EPAC1 can interact with NPAT or SUMO2/3 individually, however, the interaction between the three proteins was not detected, excluding SUMOylation as the mechanistic link between the two condensates **Extended Data Figure 7a,b**.

Together our data suggest that EPAC1 condensates can interact with other nuclear condensates, however with different modalities. The most striking difference is that between PML and NPAT. As can be seen from the superresolution images and analysis, PML and EPAC1 constitute hybrid condensates with the two proteins distributed within the same structure (**Figure 4f**). On the other hand, NPAT and EPAC1 condensates partially overlap without though constituting a single structure and maintaining their condensate "identity". These data would suggest that EPAC1 could have different regulatory activities in the cell, for instance by interaction with PML could affect SUMOylation (as suggested by Yang et al, Science Advances April 20, 2022. DOI: 10.1126/sciadv.abm2960), while on the other hand, the interaction with HLBs allows EPAC1 to impinge in histone transcription.

Minor point:

Recent work has shown that the HLB forms by phase separation (Hur et al Dev Cell 2020) and has elucidated ways to perturb this process. Addressing the effects of some of these perturbations on EPAC biocondensate could be useful.

We thank the reviewer for the suggestion. In this manuscript Hur and colleagues study the modalities of HLBs formation in *Drosophila* embryos. The authors combine bioinformatic in silico approaches with genetics on embryos, that while very interesting and informative would be too difficult and too time-consuming to apply to our manuscript. In addition, the authors also propose two pharmacological approaches that apparently blocked the recruitment of mxc (the *Drosophila* orthologue of NPAT) to the HLBs (alpha-amanitin and SNS-032), resulting in smaller HLBs. In our opinion modification of the HLB size is out of the focus of the present study.

Reviewer #3 (Remarks to the Author):

The overarching goal of this work by Iannucci et al. is to uncover the mechanism of PKA-independent nuclear cAMP signaling. To this end, the authors focus on the cAMP-activated GEF EPAC1 that is known to shuttle from the cytoplasm into the nucleus upon cAMP production. This work nicely combines mutagenesis and chemical biology to show that nuclear EPAC1 localizes to cAMP-induced and -dependent foci. The authors furthermore intriguingly link EPAC1 foci localization to differences in gene expression, most notably of histone gene clusters on chromosome 6. The authors conclude that EPAC1 forms nuclear condensates in response to PKA-independent cAMP signaling and that this cAMP/EPAC1 condensate axis represents a novel signaling paradigm. Overall, this work is interesting and thought-provoking, and in principle a nice fit for Nature Communications. However, the authors over-interpret their results throughout the manuscript, raising doubts that EPAC1 is indeed a central hub in PKA-independent cAMP signaling. Thus, this work requires major revisions. Specifically, the burden of proof has not been met for three key statements:

Major points

1.) The observed co-localization of EPAC1 Δ 179-208-YFP and RANBP2 in Extended Data Figure 1D is not sufficient to draw the conclusion that they directly bind each other, especially given that: (a) this is a novel EPAC1 mutation, (b) nuclear localization signals (NLSs) typically bind to nuclear transport receptors (NTRs) and not nucleoporins like RANBP2, and (c) there is prior biochemical evidence for a direct, NLS/NTR-independent tethering of EPAC1 to RANBP2 (Gloerich et al. 2011). Thus, to truly identify the specific mechanism of EPAC1 nuclear import (as claimed in lines 68-69), the authors must directly test binding of wild-type and mutant EPAC1 to NTRs and nucleoporins, for example by immunoprecipitation assays (IPs) or binding assays with purified components.

While we agree that the identification of the partners through which EPAC1 enters the nucleus would increase our molecular understanding on how this protein transits the nuclear envelope, we feel that this level of mechanistic detail is beyond the scope of the present study. In the field of EPAC1, at least to our knowledge, there is no unequivocal evidence to confirm that EPAC1 enters the nucleus and most literature assumes the nuclear presence of EPAC1 based on its binding to the nuclear envelope (in particular its binding to RANBP2⁶ and Nup98⁷). We reasoned that a protein as big and complex as EPAC1 would require a precise molecular mechanism to enter the nucleus, and even more importantly, the existence of a

conserved mechanism would strongly suggest a functional role of EPAC1 in that compartment.

As requested by the reviewer we performed more experiments to substantiate the data suggesting that two sequences are both important for EPAC1 entry and apparently have different roles. In particular:

1. We repeated both the Western Blotting and imaging experiments for all constructs (EPAC1-YFP, Δ 179-208 and Δ 732-764) and confirmed our previous results **Extended figure 1b-e**.
2. As requested by the reviewer we also performed pull down experiments that demonstrate the importance of residues 732-764 in the binding of RANBP2 (**Extended figure 1f**).

2.) The data presented is not sufficient to definitively conclude that EPAC1 forms condensates, as opposed to being merely recruited to condensates. In fact, the authors show evidence for the latter in the form of (a) the (incomplete) co-localization of EPAC1 foci, PML bodies and HLBs, and (b) changes in histone gene cluster transcription between wild-type and mutant EPAC1. To show that EPAC1 is indeed the core scaffold component of a novel type of condensates (that unequivocally can be called EPAC1 condensates), the authors must reconstitute the phase separation of EPAC1 in vitro.

We agree with the reviewer that reconstitution of EPAC1 condensates in vitro would further confirm the ability of EPAC1 to undergo phase separation. In a recent manuscript published while ours was under revision, Yang et al, Science Advances April 20, 2022. DOI: 10.1126/sciadv.abm2960 presented evidence for EPAC1 phase separation in vitro, however no information on EPAC1 mutants was available. For this reason, we purified EPAC1-WT and EPAC1 Δ 2-148 and performed experiments to reconstitute phase separation in vitro. We found that EPAC1WT undergoes phase separation in the presence of cAMP (in salt concentrations around 150mM which are similar to physiological ones). On the other hand, EPAC1 Δ 2-148 was insensible to cAMP variations and did not phase separate. These data are presented in **figure 3c,d** and **Extended figure 4b-d**.

3.) The author's efforts to map the domains driving the potential phase separation of EPAC1 are cursory, leaving more doubts than providing understanding for three main reasons. First, IDRs are by no means cardinal to phase separation as claimed in line 126. What is cardinal to phase separation is multivalency, which can be achieved through both folded and disordered domains. Indeed, the prevalent view in the field is that IDR typically work in concert with oligomerization domains, and that IDRs often merely tune phase separation behavior (see e.g. Martin and Holehouse, 2020). Second, EPAC1 appears to only have one very short IDR (approx. residues 1-66) based on DisoPred3, NetSurfP2 and AlphaFold2 predictions. The other IDRs that Iannucci et al. highlighted merely seem to be classic disordered linkers connecting folded domains. Third, it appears that EPAC1 contains folded domains like the DEP domain that are known to mediate protein-protein interactions in signaling pathways. In this regard, the author note that truncating the DEP domain (by virtue of the EPAC1 Δ 48-148 mutant) interferes with condensate localization. Together, the domain mapping appears biased towards pushing the preconceived model that EPAC1 undergoes IDR-dependent liquid-liquid phase separation, whereas both the data presented in this manuscript as well as the literature cast doubts on this model. Thus, the authors must

devise more comprehensive and unbiased experiments to distinguish between the EPAC1 phase separation and condensate recruitment models (and identify the domains that drive this), or significantly revise and tone down their conclusions.

As suggested by the reviewer we did both, revised and toned down our conclusions and devised new experiments that allowed us to better define the role of the amino terminus IDRs of EPAC1. The new evidence presented are:

1. In vitro experiments demonstrated that the EPAC1 amino terminus is necessary for phase separation **figure 3c,d** (please see also previous point)
2. We performed reconstitution experiments using the EPAC1 mutants and found that $\Delta 2-148$ was unable to form condensates in response to cAMP but also to participate on the condensates generated by EPAC1-WT. On the other hand, the mutants $\Delta 2-24$ and $\Delta 48-148$ that did not form condensates in response to cAMP, promptly participated in the formation of EPAC1-WT condensates suggesting a role of these domains in the processes that trigger condensation and pointing to the importance of this region both for the condensation and recruitment processes (**Extended figure 5**).

Minor points:

1. How were the EPAC1-deficient HEK cells generated? This information is missing in the manuscript, despite the importance of this resource for this study.

The HEK cell line is well documented to be naturally EPAC1 deficient⁵. The lack of EPAC is demonstrated in the manuscript in **figure 1a** by Western Blotting. In the new version of the manuscript, we make a better point on the deficiency of EPAC1 in HEK cells.

2. Lines 115-118: Imprecise argument. Hexanediol sensitivity is not diagnostic of LLPS, but merely suggests that hydrophobic interactions may be relevant. Much stronger evidence for the model that EPAC1 behaves as if in a viscous liquid phase comes from the fusion and FRAP experiments.

We thank the reviewer for the suggestion, in the new version of the manuscript we re-wrote that part accordingly.

3. It would greatly help if the authors would consolidate the EPAC1 domain architecture cartoons in Figure 3A and Extended Data Figure 1A into one scheme.

As suggested by the reviewer we have integrated the domain architecture of EPAC1 in **figure 3a**.

4. Structuring the manuscript into sections with subheadings would greatly help the flow of the paper.

The new version of the manuscript is conformed to the Nature Communications format that requires sections defined by subheadings.

Reviewer #4 (Remarks to the Author):

In this manuscript, Iannucci et al. explored the nuclear functions of cAMP effector, EPAC1. They found that upon cAMP elevation, EPAC1 enters into the nucleus where it forms reversible biomolecular condensates through liquid-liquid phase separation. This phenomenon is independent on the canonical effector of cAMP, PKA. Moreover, they found that EPAC1 condensates assemble on chromosome 6 with colocalization with histone gene regulator NPAT to promote histone gene transcription. Overall, this is an interesting story as it uncovered a mechanism through which cAMP contributes to nuclear spatial compartmentalization and promotes the transcription of specific genes. However, lacking of proper controls and highly rely on cell imaging make some conclusions not convincing. The following are my concerns that need addressed.

Major points

1. My big concern is whether and how the EPAC1 condensate regulates histone gene expression. It is well-known that histone gene expression is tightly controlled by cell cycle. It is unknown whether treatment with cAMP agonist will affect cell cycle. The authors should try to explain how EPAC1 condensates regulate histone gene expression. What about the EPAC1 mutants? Does the EPAC1 732-764 mutant and EPAC1 179-208 mutant that cannot enter into the nucleus regulate histone gene expression independent on cell cycle?

In the first version of the manuscript, we did not address the cell cycle-dependence of histone transcription because we reasoned that our treatment (40 minutes of the cell permeant EPAC-specific cAMP analog 8CPT-cAMP) was unlikely to dramatically affect the cell cycle progression. As requested by the reviewer we used Fluorescence-activated Cell Sorting (FACS) to monitor the variation of cell cycle progression in cells expressing EPAC1 treated with 8CPT-cAMP. As shown in **Extended figure 6d**, treatment with 8CPT-cAMP had no effect on cell cycle progression as compared to the vehicle control and thus is unlikely that the effects on histone transcription depend on cell cycle variations.

In regards of the effects that the different mutants may have on transcription, we feel that the results of the original RNAseq experiments, demonstrating that the Δ 2-148 mutant (which enters the nucleus, contains a fully functional catalytic domain, and can respond to cAMP but is unable to form condensates) has no transcriptional effects, clearly suggests that EPAC1 condensates are the means through which histone transcription is activated in response to cAMP. For this reason, we feel that testing the ability of EPAC1 mutants that do not enter the nucleus to alter transcription would be less informative than the Δ 2-148 mutant.

2. Fig. 4: The co-localization of NPAT with EPAC1 condensates is not that clear. Moreover, it is recommended to use other techniques, such as Co-IP to confirm the interaction between NPAT and EPAC1. ChIP EPAC1 and NPAT at histone locus is also a better way to demonstrate that point.

In the new version of the manuscript, we present new data that confirm the co-localization of NPAT and EPAC1 condensates (new **Extended figure 7a,b**). As suggested from our imaging experiments, NPAT and EPAC1 appear to be part of partially overlapping condensates

(contrary to EPAC1 and PML that are clearly part of the same structures (see also response to major point 2 Reviewer #2). As requested by the reviewer we performed co-IP experiments, but we were unable to detect interaction of EPAC1 and NPAT, which is not surprising considering the low number of interactions between HLBs and EPAC1 condensates observed by immunofluorescence (**Figure 4f**).

3. To make the story more convincing, the authors are recommended to express and purify the recombinant EPAC1 to demonstrate it can form liquid-liquid phase separation in vitro.

We now present data demonstrating the reconstruction of EPAC1 condensates in vitro (please see answer to major point 2 Reviewer #3).

4. The probe signal for chromosome 21 is very weak. It is too preliminary to conclude that there is no co-localization between chromosome 21 and EPAC1 condensates given the so weak signals.

As requested by the reviewer we repeated these experiments and obtained the same results. To further substantiate our data, we also used two new probes for the centromeric regions of chromosome 15 and 12 representative images are now presented and quantified in the new **figure 6**.

5. Extended data Fig. 1: The authors should examine whether these two domains affect the protein expression of EPAC1. For extended data 1c and 1d, WT EPAC1 should be added as a positive control.

We agree with the reviewer that the original Western Blotting data presented in **Extended figure 1** indicated a significant difference in the expression levels of the two mutants as compared to the WT protein. We repeated the Western Blotting experiments and found that the differences were exacerbated (probably due to exposure differences). In fact, by loading all the samples in the same gel we found that the two mutants are only slightly less expressed than the WT protein. Nevertheless, we believe that this slight difference in expression does not affect the conclusions of our experiments as both mutants are clearly not detectable in the nucleus. In addition, in the new version of **Extended figure 1** we added EPAC1^{WT} as control, as requested by the reviewer.

Minor points

1. Fig. 1c: what is time points used for calculation?

The treatment we used for all experiments presented is between 40 and 60 minutes.

2. Fig. 1d and 1g: The DAPI should be added to indicate the nucleus.

We added the DAPI labelling of the nucleus in all images except of the experiment presented in **Extended figure 6d-e** to facilitate the interpretation of the data (colocalization of three fluorescent proteins).

3. Fig. 5b: the EPAC1 1-148 mutant should be included to show its effect on histone gene expression.

In **Figure 5c** are depicted only the statistically significant DEGs in the comparison between cells expressing the EPAC1-148 mutant and treated with vehicle (DMSO) or 8CPT-cAMP. In

this comparison no histone gene varied significantly. As requested by the reviewer we generated the heat map of the expression of the histone genes in the three RNAseq replicates of the EPAC1-148 mutant and found no significant changes in expression. We enclose here the heat map for the reviewer's information, but we think not necessary to include this figure in the manuscript. If, however, the reviewer thinks it is necessary to include these data in the manuscript we have no objection.

4. The language of this manuscript needs to be polished. There are a lot of grammar errors and mistakes.

The manuscript has been carefully revised and formatted according to the format requested by Nature Communications.

References

1. Shio, Y. & Eisenman, R. N. Histone sumoylation is associated with transcriptional repression. *Proc. Natl. Acad. Sci. U. S. A.* **100**, 13225–13230 (2003).
2. Duronio, R. J. & Marzluff, W. F. Coordinating cell cycle-regulated histone gene expression through assembly and function of the Histone Locus Body. *RNA Biol.* **14**, 726 (2017).
3. Niimura, M. *et al.* Critical role of the N-terminal cyclic AMP-binding domain of Epac2 in its subcellular localization and function. *J. Cell. Physiol.* **219**, 652–658 (2009).
4. Schwede, F. *et al.* Structure-guided design of selective Epac1 and Epac2 agonists. *PLoS Biol.* **13**, (2015).
5. Zhu, Y., Mei, F., Luo, P. & Cheng, X. A cell-based, quantitative and isoform-specific assay for exchange proteins directly activated by cAMP. *Sci. Rep.* **7**, (2017).
6. Gloerich, M. *et al.* The nucleoporin RanBP2 tethers the cAMP effector Epac1 and inhibits its catalytic activity. *J. Cell Biol.* **193**, 1009–1020 (2011).
7. Liu, C. *et al.* The Interaction of Epac1 and Ran Promotes Rap1 Activation at the

Nuclear Envelope. *Mol. Cell. Biol.* **30**, 3956 (2010).

REVIEWERS' COMMENTS

Reviewer #1 (Remarks to the Author):

My concerns or suggestions have been taken into account and resolved in the updated version.

Reviewer #2 (Remarks to the Author):

The manuscript is improved with the new revisions and I think that the authors claims are better supported now.

Reviewer #4 (Remarks to the Author):

The authors have properly answered all my concerns and the manuscript has been improved after revision. No further concerns.

Reviewer #5 (Remarks to the Author):

Overview:

In this revised manuscript by Iannucci et al, the authors described that phase separation of nuclear EPAC1 -controlled by the second messenger cAMP- is as a bona fide signaling event that impacts transcription and potentially the function of nuclear membraneless organelles. It was found that these nuclear EPAC1-containing condensates can regulate SUMOylation, a crucial process for nuclear function, as well as the expression of a specific histone gene cluster at chromosome 6. Interestingly, the effects of nuclear EPAC1 condensates on transcription were found to be independent of SUMOylation, suggesting that they are regulated by distinct mechanisms. The study suggests that the nuclear cAMP/EPAC1-condensate axis may represent a novel molecular mechanism that could influence the physiology of cells.

Comment #1:

The authors responded to Comment #1 by claiming that investigating the molecular mechanism through which EPAC1 transits the nuclear envelope is beyond the scope of the study due to its inherent complexity. However, the authors substantiated the data by repeating all the western blotting and IF experiments for EPAC1-YFP, $\Delta 179-208$ and $\Delta 732-764$. The experimental results showed a subtle decrease in nuclear EPAC1 upon deletion of either region, while $\Delta 732-764$ further abolished the co-localization with RANBP2. These findings are further supported by the additional pulldown experiment (Extended Figure 1F), where the interaction between RANBP2 and EPAC1 is observed with the WT and $\Delta 179-208$, but not with $\Delta 732-764$. It should be noted, however, that the band quality of the immunoprecipitation is sub-optimal, possibly due to the over-expression of EPAC1-YFP. To address this, an alternative approach involving the pulldown of endogenous RANBP2 and subsequent blotting for EPAC1-YFP could be considered. Notably, in the figure (Extended Figure 1F), there are two lanes labeled EPAC1-YFP, one of which does not exhibit an interaction with RANBP2 (the rightmost lane). The authors should provide further clarification regarding this particular result.

Also, there is a typo in line 96 (nether should be neither).

Comment #2:

In addressing Comment #2, the authors purified recombinant EPAC1-WT and $\Delta 2-148$ variants, followed by conducting in vitro phase separation assays (Figures 3C-D, Extended Figures 4B-D). The findings revealed that at room temperature conditions, in the presence of cAMP, and at a salt concentration of 150 mM EPAC1-WT undergoes phase separation. However, EPAC1 $\Delta 2-148$ failed to do so and remained insensitive to changes in cAMP levels. These results demonstrate that the presence of cAMP modulates the ability of EPAC1 to phase separate in vitro, with the N-terminus of EPAC1 playing a crucial role in facilitating this process.

Comment #3:

In response to Comment #3, the authors revised their conclusions by presenting two potential models: either condensate initiation or recruitment to existing condensates. To further support their argument, the authors conducted experiments involving EPAC1 $\Delta 2-24$ and $\Delta 48-148$ variants, which demonstrated their ability to participate in the formation of condensates alongside untagged WT EPAC1 (Extended Figure 5). In contrast, $\Delta 2-148$ exhibited an inability to engage in condensate formation (Extended Figures 5C-D). The authors also successfully eliminated the dominant negative effect by co-expressing both EPA1-YFP WT and $\Delta 2-148$, wherein EPA1-YFP WT retained its capacity for phase separation in the presence of the mutant. These additional findings, combined with the results obtained from the in vitro phase separation assays, provide additional evidence supporting the assertion that EPAC1 undergoes IDR-dependent phase separation.

Minor Points:

Point #1: The authors have supported their claim that HEK cells are EPAC1-deficient at the protein level by showing western blotting data (Figure 1A) and citing existing literature.

Point #2: The authors have corrected the function of 1,6-hexanediol in the manuscript and showed fusion and FRAP results to support LLPS of EPAC1 (lines 169-179) (Figure 2).

Point #3: The authors have modified the figures accordingly (Figure 3A).

Point #4: The authors have added subheadings to improve the flow of the paper.

Overall, this reviewer thinks the authors addressed most (not all) of the raised concerns in the original submission.

Cyclic AMP-induced reversible EPAC1 condensates regulate histone transcription
Iannucci et. al.

Point-by-point response to the reviewers' comments

We would like to thank the reviewers for taking the time to review the revisited version of our manuscript. Hereafter a point-by-point response is presented.

Reviewer #1 (Remarks to the Author):

My concerns or suggestions have been taken into account and resolved in the updated version.

Thank you

Reviewer #2 (Remarks to the Author):

The manuscript is improved with the new revisions and I think that the authors claims are better supported now.

Thank you

Reviewer #4 (Remarks to the Author):

The authors have properly answered all my concerns and the manuscript has been improved after revision. No further concerns.

Thank you

Reviewer #5 (Remarks to the Author):

Overview:

In this revised manuscript by Iannucci et al, the authors described that phase separation of nuclear EPAC1 -controlled by the second messenger cAMP- is as a bona fide signaling event that impacts transcription and potentially the function of nuclear membraneless organelles. It was found that these nuclear EPAC1-containing condensates can regulate SUMOylation, a crucial process for nuclear function, as well as the expression of a specific histone gene cluster at chromosome 6. Interestingly, the effects of nuclear EPAC1 condensates on transcription were found to be independent of SUMOylation, suggesting that they are regulated by distinct mechanisms. The study suggests that the nuclear cAMP/EPAC1-condensate axis may represent a novel molecular mechanism that could influence the physiology of cells.

Comment #1:

The authors responded to Comment #1 by claiming that investigating the molecular mechanism through which EPAC1 transits the nuclear envelope is beyond the scope of the study due to its inherent complexity. However, the authors substantiated the data by

repeating all the western blotting and IF experiments for EPAC1-YFP, Δ 179-208 and Δ 732-764. The experimental results showed a subtle decrease in nuclear EPAC1 upon deletion of either region, while Δ 732-764 further abolished the co-localization with RANBP2. These findings are further supported by the additional pulldown experiment (Extended Figure 1F), where the interaction between RANBP2 and EPAC1 is observed with the WT and Δ 179-208, but not with Δ 732-764. It should be noted, however, that the band quality of the immunoprecipitation is sub-optimal, possibly due to the over-expression of EPAC1-YFP. To address this, an alternative approach involving the pulldown of endogenous RANBP2 and subsequent blotting for EPAC1-YFP could be considered. Notably, in the figure (Extended Figure 1F), there are two lanes labeled EPAC1-YFP, one of which does not exhibit an interaction with RANBP2 (the rightmost lane). The authors should provide further clarification regarding this particular result.

We agree with the reviewer that the RANBP2 bands in the Western Blotting experiment presented in **supplementary figure 1F** was sub-optimal. Unfortunately, RANBP2 is a large protein of approximately 350kD and the quality of the available antibodies is not optimal. To repeat the experiments would require the acquisition of new antibodies which will take more than the 2-weeks provided by the editor for the final revisions of the manuscript. Moreover, we reasoned that **supplementary figure 1f** was used to confirm that the C-terminus of EPAC1 contains the sequences necessary for engaging RANBP2. However, this is well accepted and has been demonstrated previously. For these reasons, in the final version of the manuscript we decide NOT to include the panel f of Supplementary figure 1. We therefore removed the description of this panel from the main manuscript and added the references demonstrating that EPAC1 can interact with RANBP2.

Also, there is a typo in line 96 (nether should be neither).

Corrected

Comment #2:

In addressing Comment #2, the authors purified recombinant EPAC1-WT and Δ 2-148 variants, followed by conducting in vitro phase separation assays (Figures 3C-D, Extended Figures 4B-D). The findings revealed that at room temperature conditions, in the presence of cAMP, and at a salt concentration of 150 mM EPAC1-WT undergoes phase separation. However, EPAC1 Δ 2-148 failed to do so and remained insensitive to changes in cAMP levels. These results demonstrate that the presence of cAMP modulates the ability of EPAC1 to phase separate in vitro, with the N-terminus of EPAC1 playing a crucial role in facilitating this process.

Comment #3:

In response to Comment #3, the authors revised their conclusions by presenting two potential models: either condensate initiation or recruitment to existing condensates. To further support their argument, the authors conducted experiments involving EPAC1 Δ 2-24 and Δ 48-148 variants, which demonstrated their ability to participate in the formation of condensates alongside untagged WT EPAC1 (Extended Figure 5). In contrast, Δ 2-148 exhibited an inability to engage in condensate formation (Extended Figures 5C-D). The authors also successfully eliminated the dominant negative effect by co-expressing both EPA1-YFP WT and Δ 2-148, wherein EPA1-YFP WT retained its capacity for phase separation in the presence of the mutant. These additional findings, combined with the results obtained from the in vitro phase

separation assays, provide additional evidence supporting the assertion that EPAC1 undergoes IDR-dependent phase separation.

Minor Points:

Point #1: The authors have supported their claim that HEK cells are EPAC1-deficient at the protein level by showing western blotting data (Figure 1A) and citing existing literature.

Point #2: The authors have corrected the function of 1,6-hexanediol in the manuscript and showed fusion and FRAP results to support LLPS of EPAC1 (lines 169-179) (Figure 2).

Point #3: The authors have modified the figures accordingly (Figure 3A).

Point #4: The authors have added subheadings to improve the flow of the paper.

Overall, this reviewer thinks the authors addressed most (not all) of the raised concerns in the original submission.